# PIORF: Physics-Informed Ollivier–Ricci Flow for Long-Range Interactions in Mesh Graph Neural Networks

**Youn-Yeol Yu**[1]*✉, **Jeongwhan Choi**[1]*✉, **Jaehyeon Park**[2]✉, **Kookjin Lee**[3]✉, **Noseong Park**[2]†✉
[1]Yonsei University    [2]KAIST    [3]Arizona State University

## Abstract

Recently, data-driven simulators based on graph neural networks have gained attention in modeling physical systems on unstructured meshes. However, they struggle with long-range dependencies in fluid flows, particularly in refined mesh regions. This challenge, known as the 'over-squashing' problem, hinders information propagation. While existing graph rewiring methods address this issue to some extent, they only consider graph topology, overlooking the underlying physical phenomena. We propose Physics-Informed Ollivier–Ricci Flow (PIORF), a novel rewiring method that combines physical correlations with graph topology. PIORF uses Ollivier–Ricci curvature (ORC) to identify bottleneck regions and connects these areas with nodes in high-velocity gradient nodes, enabling long-range interactions and mitigating over-squashing. Our approach is computationally efficient in rewiring edges and can scale to larger simulations. Experimental results on 3 fluid dynamics benchmark datasets show that PIORF consistently outperforms baseline models and existing rewiring methods, achieving up to 26.2% improvement.

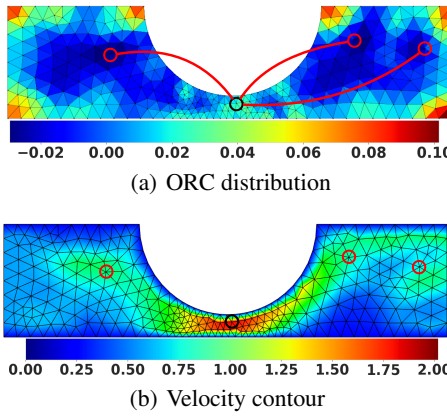

(a) ORC distribution

(b) Velocity contour

Figure 1: Visualization of PIORF rewiring in CylinderFlow-Tiny. (a) Blue areas indicate potential bottlenecks. Red circles (◯) denote critical bottleneck nodes. (b) The black circle (◯) denotes the highest velocity node. PIORF connects bottleneck nodes (◯) with high-velocity nodes (◯).

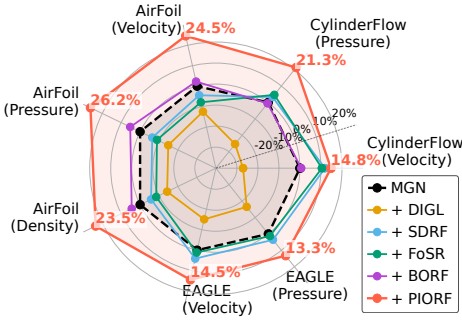

Figure 2: The radar plot shows the percentage improvement over MGN for each method on 3 datasets. The radial distance indicates the magnitude of improvement. PIORF consistently outperforms other methods with substantial gains particularly in AirFoil (24.5% for Velocity) and CylinderFlow (21.3% for Pressure).

## 1 Introduction

Solving the Navier–Stokes equations that govern fluid dynamics remains an open problem. In the absence of an analytical solution, most studies use numerical methods, representatively, finite ele-

---

*Equal contribution.
†Corresponding author.

ment methods (FEMs) (Madenci & Guven, 2015; Stolarski et al., 2018; Abaqus, 2011; Dhatt et al., 2012) to discretize differential equations spatially and temporally to account for complex physics. To optimize computational resources while maintaining accuracy in simulations involving unstructured surfaces, mesh refinement techniques are commonly used. These methods allocate higher resolution to regions of interest that require more detailed analysis, such as areas with steep gradients or complex geometries. While this approach balances computational cost with simulation accuracy, it results in a complex and irregular mesh structure (Löhner, 1995; Liu et al., 2022).

The high computational cost of traditional numerical solutions has sparked interest in data-driven simulators based on graph neural networks (GNNs). Graph machine learning approaches, particularly MeshGraphNets (MGNs) (Pfaff et al., 2020), have shown promising results in modeling physical systems on unstructured meshes. So far, studies using MGNs have shown accurate predictions for various physical systems (Sanchez-Gonzalez et al., 2020; Fortunato et al., 2022; Yu et al., 2024). However, these methods face the challenge of capturing the long-range dependence of fluid flows, which is essential for accurately simulating complex phenomena such as turbulence (Benzi & Toschi, 2023).

**Mesh refinement and over-squashing problem.** The core problem in using GNNs for fluid dynamics simulations lies in balancing mesh refinement and information propagation. To achieve accurate simulations, it is essential to use finer meshes, especially in regions with significant velocity gradients, such as in boundary conditions (e.g. walls, holes, inlets, and outlets) (Katz & Sankaran, 2011; Baker, 2005). However, this refinement introduces two critical issues: i) as information propagates through the graph, it is repeatedly compressed, leading to an 'over-squashing' problem (Alon & Yahav, 2021; Topping et al., 2021). The over-squashing occurs in areas of local mesh refinement (Imai & Aoki, 2006) and near boundary conditions where the mesh is non-uniform, resulting in some nodes having few neighbors. ii) As the mesh becomes finer, MGNs need to perform more message-passing steps to propagate information over the same physical distance. This leads to 'under-reaching' problems (Fortunato et al., 2022), where the model struggles to capture interactions beyond a certain range. These issues are particularly pronounced in fluid dynamics simulations. As the mesh becomes finer, the challenges increase, creating a trade-off between the demand for high-resolution simulations and the capacity of GNNs to efficiently process the graphs.

**Limitations of existing solutions.** While several graph rewiring methods have been proposed to address over-squashing (Topping et al., 2021; Karhadkar et al., 2022; Nguyen et al., 2023; Black et al., 2023; Arnaiz-Rodríguez et al., 2022), they typically consider only the graph topology. This approach is insufficient for fluid dynamics simulations, where the underlying physical phenomena play a crucial role in determining important long-range interactions.

**Main idea.** To address these challenges, we propose Physics-Informed Ollivier–Ricci Flow (PI-ORF)[1], a novel method that incorporates physical quantities such as flow velocity into graph rewiring. PIORF uses graph topology and physical phenomena to reduce over-squashing and enhance information flow. We use the Ollivier–Ricci curvature (ORC) (Ollivier, 2009) to identify bottleneck regions in the graph structure. Fig. 1 depicts the key idea behind our PIORF using a CYLINDERFLOW-TINY simulation. The ORC distribution (Fig. 1(a)) reveals potential bottleneck areas (blue regions), with red circles (○) marking nodes of minimum curvature. The velocity magnitude contour (Fig. 1(b)) shows areas of rapid fluid velocity changes, with the black circle (○) indicating the highest velocity node. Our approach connects these bottleneck nodes with nodes in high-velocity gradient regions, enabling long-range interactions and mitigating over-squashing.

**Contributions.** Our contributions are summarized as follows:

- To the best of our knowledge, we are the first to introduce a rewiring method that considers both graph topology and physical phenomena for fluid dynamics simulations.
- Our PIORF method shows excellent computational efficiency by adding multiple edges with a single calculation compared to existing rewiring methods.
- We extend PIORF to handle temporal mesh graphs and apply it to dynamic simulation environments such as the EAGLE dataset, demonstrating the scalability of PIORF to larger mesh graphs.

---

[1]Our code is available here: https://github.com/yuyudeep/piorf

- As shown in Fig. 2, PIORF consistently outperforms MGN model and other rewiring methods across 3 benchmark datasets, achieving up to 26.2% improvement.

## 2 RELATED WORK

### 2.1 MESH-BASED SIMULATION MODELS

Using GNNs to predict the results of complex physical systems is a popular area of scientific machine learning (SciML) (Li et al., 2020; Michałowska et al., 2023; Belbute-Peres et al., 2020; Mrowca et al., 2018; Li et al., 2019; 2018; Pfaff et al., 2020). Among them, MGN performs local message passing by re-expressing it as a graph from a mesh. The strength of MGN lies in its ability to use mesh-based representations commonly used in many commercial simulation tools to numerically solve partial differential equations (PDEs). Instead of solving the PDEs directly, MGN learns the underlying dynamics from data and can be applied to a variety of systems while incorporating boundary conditions. However, in order to obtain a more accurate solution approximate, MGN often requires finer meshes. A larger number of nodes causes the GNN's under-reaching problem and requires more layers for effective long-range interactions, which reduces learning efficiency. To address this, recent studies have investigated methods to enable long-range interaction by forming a hierarchical structure (Fortunato et al., 2022; Cao et al., 2023) or using a Transformer (Janny et al., 2023; Yu et al., 2024). Fortunato et al. (2022) introduce a dual-layer structure designed to propagate messages at two different resolutions. Janny et al. (2023) proposes a clustering-based pooling method and performs global self-attention. Cao et al. (2023) reviews the shortcomings of current pooling methods and proposes Bi-Stride Multi-Scale (BSMS), a hierarchical GNN using bi-stride pooling. Yu et al. (2024) use hierarchical mesh graphs and has an ability to capture long-range interactions between spatially distant locations within an object.

### 2.2 OVER-SQUASHING AND GRAPH REWIRING METHODS

The issue of over-squashing was initially identified by Alon & Yahav (2021) and has since emerged as a significant challenge in GNNs when dealing with long-range dependencies. This phenomenon occurs when the information aggregated from a large number of neighbors is compressed into a fixed-sized node feature vector, resulting in a considerable loss of information (Alon & Yahav, 2021). Several approaches have been studied to address the over-squashing problem in GNNs (Finkelshtein et al., 2023; Shi et al., 2023; Errica et al., 2023; Choi et al., 2024; Fesser & Weber, 2024; Choi et al., 2025). While alternative message-passing strategies, such as expanded width-aware message passing (Choi et al., 2024), have gained attention, graph rewiring – adding or removing edges – has been the most actively proposed (Gasteiger et al., 2019; Topping et al., 2021; Nguyen et al., 2023; Arnaiz-Rodríguez et al., 2022; Karhadkar et al., 2022; Black et al., 2023; Banerjee et al., 2022; Attali et al., 2024). Gasteiger et al. (2019) propose DIGL rewiring method that computes kernel evaluation and sparsification of the adjacency matrix. DIGL smooths the adjacency of the graph, which makes it tend to connect nodes at short distances (Coifman & Lafon, 2006). However, this makes it not suitable for tasks that require longer diffusion distances. Topping et al. (2021) propose a curvature-based graph rewiring strategy. This method identifies edges with minimal negative curvature and adds new edges around them. First-order spectral rewiring (FoSR) proposed by Karhadkar et al. (2022) calculates the change in spectral gap due to edge addition and selects the edge that maximizes the gap. Nguyen et al. (2023) propose batch Ollivier–Ricci flow (BORF) using ORC to simultaneously solve the over-smoothing and over-squashing problems. BORF works in batches and calculates the curvature with a minimum and maximum in each batch. Then, connections are added to the set with the minimum edge value to uniformly weaken the graph bottleneck. BORF does not recalculate the graph curvature within each batch, but rather reuses the already computed optimal transfer plan between sets to determine which edges should be added. Recently, Attali et al. (2024) alleviate over-squashing by using Delaunay triangulation, but this is not appropriate because mesh-based simulations are already constructed by the triangulation.

Despite interest in over-squashing in GNNs, over-squashing in mesh-based GNNs such as MGN remains unexplored (See Table 5). Since mesh structures have different characteristics from graph structures used in existing research, and existing rewiring methods define bottlenecks for graph topologies from a geometric perspective, it is necessary to verify that existing rewiring methods are suitable for mesh graphs with a certain number of distributed edges.

## 3 PRELIMINARIES

### 3.1 MESHGRAPHNETS (MGN)

MGNs (Pfaff et al., 2020) are a class of GNNs designed for mesh-based simulation, using an Encoder-Processor-Decoder framework. The encoder encodes as multigraph, the nodes of the mesh are converted to graph nodes, and the mesh edges become bidirectional mesh-edges. The processor updates all node and edge embeddings by performing multiple message passing along the mesh edges through multiple GraphNet blocks (Sanchez-Gonzalez et al., 2020). Finally, the decoder predicts the subsequent state by using the updated latent node representations.

**Encoder.** The mesh $\mathcal{M}^t$ at time $t$ is transformed into a graph $\mathcal{G} = (\mathcal{V}, \mathcal{E})$, where the mesh nodes become graph nodes $v_i \in \mathcal{V}$, and the mesh edges become bidirectional edges $(i, j) \in \mathcal{E}$. For each edge, we define the mesh edge feature $\mathbf{m}_{ij}$, which encodes connectivity information. The edge features are derived from the relative displacement vector $\mathbf{x}_{ij} = \mathbf{x}_i - \mathbf{x}_j$ and its norm $|\mathbf{x}_{ij}|$. Node features include the velocity $\mathbf{w}_i$ and the node type $\mathbf{n}_i$, which indicates the boundary conditions. The input and output characteristics for each dataset are detailed in Appendix C.2.

**Processor.** The processor consists of several GraphNet blocks. Each block sequentially updates node and edge embeddings through message passing operations. $\mathbf{v}_i^l$ and $\mathbf{e}_{ij}^l$ denote the node and edge embeddings at layer $l$, respectively. The update equations are:

$$\mathbf{e}_{ij}^{l+1} = f_E(\mathbf{e}_{ij}^l, \mathbf{v}_i^l, \mathbf{v}_j^l), \quad \mathbf{v}_i^{l+1} = f_V\left(\mathbf{v}_i^l, \sum_{j \in \mathcal{N}_i} \mathbf{e}_{ij}^{l+1}\right), \tag{1}$$

where $f_E$ and $f_V$ are learnable functions parameterized as multi-layer perceptrons (MLPs), and $\mathcal{N}_i$ denotes the set of neighbors of node $i$.

**Decoder and updater.** To predict the next time state from the current time, an MLP decoder is used to predict one or more output features $\mathbf{o}_i$, such as the velocity gradient $\hat{\dot{\mathbf{w}}}_i$, density gradient $\hat{\dot{\rho}}_i$ and the next pressure $\hat{p}_i$. The velocity gradient is used to calculate the next velocity $\hat{\mathbf{w}}_i^{t+1}$ through an updater, which performs a first-order integration ($\hat{\mathbf{w}}_i^{t+1} = \hat{\dot{\mathbf{w}}}_i^t + \mathbf{w}_i^t$).

**Training loss.** Following the MGN approach, the training loss uses the mean squared error (MSE):

$$\mathcal{L} = \frac{1}{|\mathcal{V}|} \sum_{i=1}^{|\mathcal{V}|} (\mathbf{w}_i^{t+1} - \hat{\mathbf{w}}_i^{t+1})^2 + \frac{1}{|\mathcal{V}|} \sum_{i=1}^{|\mathcal{V}|} (p_i^{t+1} - \hat{p}_i^{t+1})^2, \tag{2}$$

where $|\mathcal{V}|$ is the number of nodes.

### 3.2 OLLIVIER–RICCI CURVATURE ON GRAPHS

Ricci curvature, a fundamental concept in differential geometry, describes the average dispersion of geodesics in the local region of a Riemannian manifold. In the context of graphs, ORC (Ollivier, 2009) extends these concepts to graphs and considers random walks between nearby points using Wasserstein distances between Markov chains.

Given a graph $G = (\mathcal{V}, \mathcal{E})$ and a pair of nodes $i, j \in \mathcal{V}$, ORC $\kappa(i, j)$ of edge $(i, j) \in \mathcal{E}$ is defined as:

$$\kappa(i, j) = 1 - \frac{W_1(\mathbf{m}_i, \mathbf{m}_j)}{d(i, j)}, \tag{3}$$

where $d(i, j)$ is the shortest-path distance between nodes $i$ and $j$, $\mathbf{m}_i$ is probability distribution of 1-step random walk from node $i$, and $W_1$ is the Wasserstein distance of order 1. For a node $p \in \mathcal{V}$, $\mathbf{m}_i(p)$ represents the probability that a random walker starting at $i$ will reach $p$ in one step. The Wasserstein distance $W_1(\mathbf{m}_i, \mathbf{m}_j)$ between probability distributions $\mathbf{m}_i$ and $\mathbf{m}_j$ is defined as:

$$W_1(\mathbf{m}_i, \mathbf{m}_j) = \inf_{\pi \in \Pi(\mathbf{m}_i, \mathbf{m}_j)} \left( \sum_{(p,q) \in \mathcal{V}^2} \pi(p, q) d(p, q) \right), \tag{4}$$

where $\Pi(\mathbf{m}_i, \mathbf{m}_j)$ is the set of joint probability distributions with marginals $\mathbf{m}_i$ and $\mathbf{m}_j$.

ORC quantifies the variance of a geodesic and has positive, negative, and zero values. When it is 0 ($\kappa(i,j) = 0$), the geodesics tend to remain parallel, when it is a negative value ($\kappa(i,j) < 0$), they diverge, and when it is a positive value ($\kappa(i,j) > 0$), they converge. ORC on edges with high negative values is known to cause over-squashing (Topping et al., 2021). Equation (4) requires defining the probability distribution for function neighbor nodes. Since the radius of the neighbor nodes in the graph is 1, a given one-step random walk $\mathbf{m}$ from node $i$ to node $p$ is defined as:

$$\mathbf{m}_i(p) = \begin{cases} \frac{1}{\deg(i)} & \text{if } p \in \mathcal{N}_i, \\ 0 & \text{otherwise}, \end{cases} \tag{5}$$

where $\deg(i)$ is the degree of node $i$, which means the number of element in $\mathcal{N}_i$.

## 4 PIORF: PHYSICS-INFORMED OLLIVIER–RICCI FLOW

In this section, we introduce PIORF, a novel rewiring method to improve long-range dependencies.

**Design Goals.** Our proposed method is designed with the following 3 goals:

- (*Physical Context*) The method should incorporate physical quantities (e.g., velocity) with topology (e.g., ORC) to improve long-range interactions.
- (*Efficiency*) The computational cost of adding new edges should be lower than that of existing rewiring methods.
- (*Accuracy*) The prediction error should be lower than that of other rewiring methods.

**Rewiring with PIORF.** To achieve these goals, PIORF selects nodes based on their topological properties and physical quantities. We extend the ORC to node-level curvature, denoted as $\gamma_i$ for a node $i$. This node curvature $\gamma_i$ is computed as:

$$\gamma_i = \frac{1}{|\mathcal{N}_i|} \sum_{j \in \mathcal{N}_i} \kappa(i,j). \tag{6}$$

The rewiring proceeds as Algorithm 1, of which the key steps are described as follows:

i) PIORF selects $\lfloor \delta|\mathcal{V}| \rfloor$ nodes with the lowest $\gamma_i$ from $\mathcal{V}$ in order to form a set $S$, so that $|S| = \lfloor \delta|\mathcal{V}| \rfloor$ where $\delta \in (0,1)$ is the pooling ratio. $\delta$ is our *sole hyperparameter*.

ii) For each $s \in S$, PIORF computes the Euclidean distance $d(w_s, w_i)$ between velocities $w_s$ and $w_i$ for all nodes $i \in \mathcal{V} \setminus s$.

iii) For each $s \in S$, PIORF identifies nodes $r = \arg\max_{i \in \mathcal{V} \setminus s} d(w_s, w_i)$ with the largest velocity differences and defines their set as $R_s$.

iv) PIORF adds bidirectional edges $(s, r)$ and $(r, s)$ to the graph $\mathcal{G}$ for all $s \in S$ and $r \in R_s$.

For the sake of convenience in explanation, the physical quantity used in PIORF is described based on the use of velocity from the node features. By integrating both physical and topological properties, PIORF enhances long-range interactions and mitigates over-squashing. The detailed description of the notations used in the formulas written so far is summarized in Appendix E.

---

**Algorithm 1** Physics-Informed Ollivier–Ricci Flow (PIORF)

1: **Input:** A graph $\mathcal{G} = (\mathcal{V}, \mathcal{E})$, pooling ratio $\delta \in (0,1)$, velocity $w_i$ of node $i$
2: **Output:** Rewired graph $\mathcal{G}' = (\mathcal{V}, \mathcal{E}')$
3: Calculate the ORC, $\gamma_i$, for all nodes $i$ in $\mathcal{V}$ using Equation (6)
4: Selects $\lfloor \delta|\mathcal{V}| \rfloor$ nodes with the lowest $\gamma_i$ in order to form a set $S$, where $|S| = \lfloor \delta|\mathcal{V}| \rfloor$.
5: For each node $s \in S$, calculate the Euclidean distance $d(w_s, w_i)$ between velocities of $s$ and all other nodes $i \in \mathcal{V} \setminus s$.
6: Find the node $r = \arg\max_{i \in \mathcal{V} \setminus s} d(w_s, w_i)$ with the largest velocity differences.
7: Add bidirectional edges $(s, r)$ and $(r, s)$ to $\mathcal{E}'$.
8: **return** $\mathcal{G}' = (\mathcal{V}, \mathcal{E}')$

---

**Physical interpretation.** In fluid dynamics, the distinction between laminar (Schubauer & Skramstad, 1947) and turbulent (Mathieu & Scott, 2000) flows, as quantified by velocity and the Reynolds numbers (Lissaman, 1983), is important for understanding system behavior. The relationship between velocity and pressure is described through the rate of change and is explained by the Navier-Stokes equations (Temam, 2001). The velocity refers to the speed at which a fluid moves at a specific point in space. The pressure is the force exerted by a fluid per unit area on the surfaces. PIORF integrates this physical context by adding edges between nodes with significant velocity differences. This allows the model to help with long-range interactions to better simulate real-world phenomena such as fluid turbulence. Physically, connecting nodes with large velocity differences indicates regions of instability. Unlike existing rewiring methods, PIORF ensures that the rewiring process takes on the actual physical context of the system, leading to physically meaningful signal propagation. With this physical insight, PIORF can improve long-range interactions and prediction performance in physics-based simulations.

**Computational efficiency.** Unlike existing rewiring methods, which rely on greedy algorithms to iteratively add edges based on their objective functions (Karhadkar et al., 2022; Black et al., 2023), PIORF introduces a more efficient approach. PIORF identifies nodes with significant differences in physical quantities and adds new edges in a single pass. This avoids the high computational cost of iterative edge addition and, thus, improves scalability. We show that PIORF is more efficient than other rewiring methods in rewiring new edges in Section 6.3.

## 5 DISCUSSION

In this section, we analyze mesh graphs and distinguish our method from graph pooling techniques.

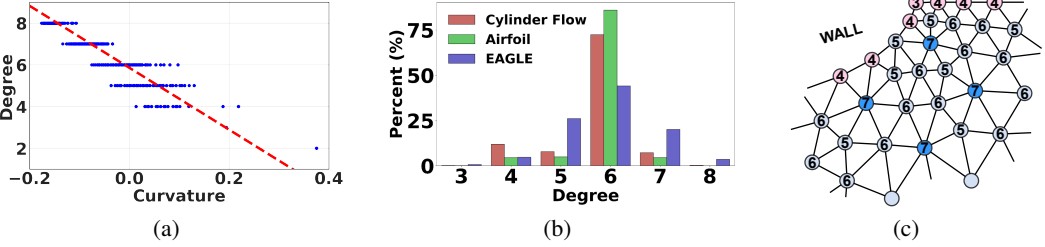

Figure 3: Structural analyses of mesh graphs: (a) Correlation between ORC and node degree in training dataset of CYLINDERFLOW, revealing potential information bottlenecks. (b) Node degree distribution across datasets, showing the prevalence of degree-6 nodes in uniform regions. (c) Non-uniform mesh refinement near boundary conditions.

**Analysis of mesh graphs.** We use ORC to analyze the topology of mesh graphs of fluid dynamics benchmark datasets. This analysis reveals several key insights:

- Fig. 3(a) shows a strong negative correlation between ORC and node degree. This relationship identifies potential information bottlenecks in the mesh graph, particularly in high-degree nodes.
- Fig. 3(b) indicates a prevalence of degree-6 nodes in uniform regions, typical of Delaunay triangulation (Weatherill, 1992). However, boundary condition nodes (e.g., holes, walls, inlets, and outlets) show lower degrees due to their sparse distribution, as shown in Fig. 3(c).
- In computational fluid dynamics (CFD) (Anderson & Wendt, 1995), local mesh refinement is often applied to enhance accuracy in specific areas. This process leads to a gradual transition from fine meshes near boundaries to coarser meshes, resulting in non-uniform structures (Fig. 3(c)).

These findings emphasize the relationship between the mesh configuration, boundary conditions, and the risk of information bottlenecks in GNNs used for fluid dynamics simulations. Fig. 8 in Appendix B shows the ORC distribution obtained through this information for each dataset.

**Pooling and rewiring methods in mesh graphs.** Due to mesh graphs with more than thousands of nodes, node pooling techniques (Fortunato et al., 2022; Cao et al., 2023; Yu et al., 2024) are widely

used to reduce computational complexity and enhance the capture of long-range interactions. We extend the application of our PIORF beyond MGN to hierarchical models such as BSMS (Cao et al., 2023) and HMT (Yu et al., 2024). While these models already incorporate pooling to effectively reduce the number of nodes, we hypothesize that applying PIORF to the pooled structures could further optimize edge connections. This integration of pooling and rewiring aims to refine capacity of the model to represent complex physical relationships across different scales. In Section 6, we explore whether this combination can yield additional improvements in fluid dynamics benchmarks.

## 6 EXPERIMENTS

### 6.1 EXPERIMENTS ON FLUID DYNAMICS BENCHMARK DATASETS

**Datasets.** We evaluate our method on two publicly available datasets: CYLINDERFLOW and AIRFOIL. Both datasets follow the Navies–Stokes equations (Temam, 2001), but differ in their flow characteristics. CYLINDERFLOW shows laminar flow behavior. In contrast, AIRFOIL represents a turbulent flow model with high velocity, where fluid particles move irregularly in time and space. (See Appendix B for a detailed description of datasets.)

**Setting.** We compare our PIORF against rewiring methods: DIGL (Gasteiger et al., 2019), FoSR (Karhadkar et al., 2022), SDRF (Topping et al., 2021), and BORF (Nguyen et al., 2023) These are apllied to 4 different models architectures: MGN (Pfaff et al., 2020), BSMS (Cao et al., 2023), Graph Transformer (GT) (Dwivedi & Bresson, 2020) and HMT (Yu et al., 2024). BSMS is a hierarchical GNN and HMT is a hierarchical Transformer. For MGN, we use 15 blocks. For optimal performance, BSMS is set to level 7 for CYLINDERFLOW and level 9 for AIRFOIL. Detailed hyperparameters for all baselines are provided in Appendix C. Each experiment is repeated 5 times with different random seeds. All experiments are performed on NVIDIA 3090 and Intel Core-i9 CPUs.

**Results.** Table 1 shows a comprehensive performance comparison of different rewiring methods across the 4 model architectures. PIORF consistently outperforms other rewiring baselines when applied to MGN, BSMS, GT, and HMT models. For CYLINDERFLOW, PIORF achieves the lowest RMSE in both velocity and pressure when applied to MGN. This improvement is especially significant compared to MGN and other rewiring methods. For AIRFOIL, PIORF achieves the best performance in all cases. Fig. 4 shows the superiority of PIORF by showing velocity magnitude contours at the final timestep. Our PIORF results closely align with the ground truth, especially in regions marked by black boxes.

**Sensitivity to pooling ratio.** We analyze sensitivity to pooling ratio $\delta$, which is *our sole hyperparameter* and determines the number of new edge connections. Fig. 5 shows how rollout RMSE varies with $\delta$ for both datasets. Velocity RMSE of CYLINDERFLOW is optimal at 3%, while pressure RMSE generally improves with higher ratios. For AIRFOIL, velocity RMSE is best at 7%, and pressure RMSE at 7%. Across all cases, 1% pooling ratio often performs worse than MGN, while 9% increases standard deviations. The results show the need to tune $\delta$ specfically for each dataset. Fig. 5 (a) shows the Velocity RMSE of CYLINDERFLOW, and it can be seen that the average and standard deviation of RMSE increase at 9% where a large number of edges are connected.

### 6.2 SCALING TO LARGER FLUID DYNAMICS

**Datasets.** To evaluate scalability and efficiency of PIORF, we use EAGLE (Janny et al., 2023), which simulates turbulent flows created by drones in various scenes. As shown in Table 2, EAGLE significantly surpasses CYLINDERFLOW and AIRFOIL in

Table 2: Comparison of fluid dynamics datasets

| Dataset | Size | Dynamic Scene | Dynamic Mesh |
|---|---|---|---|
| CYLINDERFLOW | 15GB | ✗ | ✗ |
| AIRFOIL | 56GB | ✗ | ✗ |
| EAGLE | 270GB | ✓ | ✓ |

scale and complexity. EAGLE has dynamic meshes (Malcevic & Ghattas, 2002; Jasak, 2009), where the mesh positions and boundary conditions change at each time step. This dynamic nature requires temporal graph rewiring, presenting a more challenging and realistic scenario compared to the static meshes of CYLINDERFLOW and AIRFOIL.

Table 1: RMSE (rollout-all, $\times 10^3$) for our PIORF and other rewiring methods.

| Method | CYLINDERFLOW | | AIRFOIL | | |
|---|---|---|---|---|---|
| | Velocity | Pressure | Velocity | Pressure | Density |
| MGN | $48.8 \pm 5.6$ | $36.7 \pm 2.4$ | $10{,}261 \pm 832$ | $3{,}043{,}186 \pm 282{,}514$ | $29.4 \pm 2.7$ |
| + DIGL | $62.0 \pm 1.7$ | $46.0 \pm 0.4$ | $11{,}534 \pm 623$ | $3{,}495{,}260 \pm 252{,}832$ | $33.6 \pm 2.2$ |
| + SDRF | $43.0 \pm 3.0$ | $35.5 \pm 1.0$ | $10{,}714 \pm 669$ | $3{,}238{,}730 \pm 183{,}094$ | $31.1 \pm 1.9$ |
| + FoSR | $43.7 \pm 3.2$ | $35.0 \pm 1.2$ | $11{,}068 \pm 377$ | $3{,}314{,}506 \pm 164{,}026$ | $31.9 \pm 1.5$ |
| + BORF | $48.5 \pm 7.8$ | $36.9 \pm 2.2$ | $10{,}029 \pm 410$ | $2{,}884{,}555 \pm 186{,}003$ | $28.1 \pm 1.8$ |
| + PIORF | $\mathbf{41.6} \pm \mathbf{3.9}$ | $\mathbf{28.9} \pm \mathbf{1.5}$ | $\mathbf{7{,}743} \pm \mathbf{584}$ | $\mathbf{2{,}245{,}858} \pm \mathbf{142{,}452}$ | $\mathbf{22.5} \pm \mathbf{1.4}$ |
| BSMS | $78.7 \pm 2.8$ | $50.7 \pm 2.2$ | $10{,}883 \pm 460$ | $2{,}640{,}398 \pm 158{,}480$ | $26.5 \pm 2.1$ |
| + DIGL | $237.8 \pm 7.3$ | $163.6 \pm 8.5$ | $40{,}312 \pm 3{,}936$ | $8{,}218{,}660 \pm 1{,}281{,}200$ | $81.3 \pm 11.5$ |
| + SDRF | $78.0 \pm 4.1$ | $50.7 \pm 1.9$ | $36{,}539 \pm 3{,}980$ | $7{,}426{,}023 \pm 642{,}555$ | $74.2 \pm 8.1$ |
| + FoSR | $82.2 \pm 3.8$ | $52.3 \pm 3.0$ | $41{,}831 \pm 2{,}011$ | $8{,}490{,}283 \pm 352{,}622$ | $84.0 \pm 2.4$ |
| + BORF | $84.9 \pm 2.3$ | $54.2 \pm 1.6$ | $10{,}750 \pm 430$ | $2{,}632{,}487 \pm 126{,}177$ | $25.8 \pm 1.2$ |
| + PIORF | $\mathbf{76.9} \pm \mathbf{3.8}$ | $\mathbf{50.6} \pm \mathbf{2.4}$ | $\mathbf{10{,}482} \pm \mathbf{500}$ | $\mathbf{2{,}584{,}690} \pm \mathbf{163{,}680}$ | $\mathbf{25.4} \pm \mathbf{1.6}$ |
| GT | $54.3 \pm 7.3$ | $40.0 \pm 2.0$ | $10{,}002 \pm 218$ | $2{,}979{,}573 \pm 99{,}293$ | $29.0 \pm 0.9$ |
| + DIGL | $68.4 \pm 3.6$ | $49.5 \pm 1.0$ | $11{,}004 \pm 511$ | $3{,}331{,}160 \pm 192{,}098$ | $32.7 \pm 1.9$ |
| + SDRF | $52.1 \pm 9.1$ | $39.0 \pm 1.3$ | $10{,}354 \pm 610$ | $3{,}120{,}743 \pm 179{,}587$ | $30.1 \pm 1.9$ |
| + FoSR | $50.7 \pm 8.7$ | $39.3 \pm 1.6$ | $11{,}211 \pm 868$ | $3{,}415{,}094 \pm 312{,}517$ | $33.6 \pm 3.1$ |
| + BORF | $58.9 \pm 9.7$ | $40.6 \pm 2.4$ | $9{,}830 \pm 416$ | $2{,}883{,}648 \pm 136{,}064$ | $28.6 \pm 1.3$ |
| + PIORF | $\mathbf{48.5} \pm \mathbf{4.5}$ | $\mathbf{31.3} \pm \mathbf{2.3}$ | $\mathbf{7{,}429} \pm \mathbf{778}$ | $\mathbf{2{,}124{,}920} \pm \mathbf{130{,}279}$ | $\mathbf{21.4} \pm \mathbf{1.2}$ |
| HMT | $71.0 \pm 1.2$ | $51.1 \pm 1.5$ | $5{,}303 \pm 414$ | $1{,}251{,}955 \pm 79{,}764$ | $12.8 \pm 0.8$ |
| + DIGL | $76.3 \pm 1.7$ | $54.0 \pm 0.7$ | $5{,}176 \pm 409$ | $1{,}232{,}486 \pm 79{,}250$ | $12.5 \pm 0.8$ |
| + SDRF | $71.0 \pm 1.0$ | $51.3 \pm 0.6$ | $32{,}695 \pm 1{,}013$ | $7{,}579{,}699 \pm 247{,}004$ | $74.4 \pm 2.3$ |
| + FoSR | $72.1 \pm 1.5$ | $52.3 \pm 1.1$ | $35{,}474 \pm 1{,}011$ | $8{,}137{,}115 \pm 231{,}038$ | $79.4 \pm 2.0$ |
| + BORF | $74.2 \pm 3.1$ | $53.6 \pm 1.2$ | $5{,}591 \pm 416$ | $1{,}306{,}555 \pm 82{,}509$ | $13.3 \pm 0.8$ |
| + PIORF | $\mathbf{70.9} \pm \mathbf{1.6}$ | $\mathbf{50.9} \pm \mathbf{0.8}$ | $\mathbf{4{,}961} \pm \mathbf{378}$ | $\mathbf{1{,}182{,}495} \pm \mathbf{67{,}499}$ | $\mathbf{12.1} \pm \mathbf{0.7}$ |

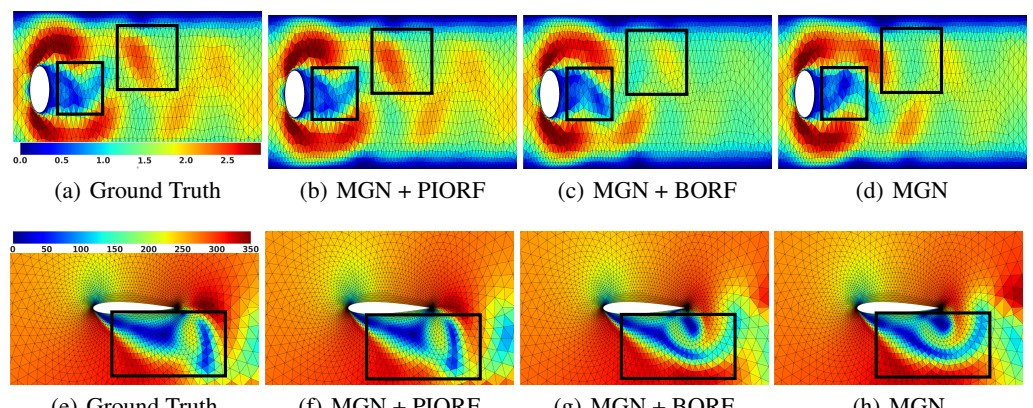

| | | | |
|---|---|---|---|
| (a) Ground Truth | (b) MGN + PIORF | (c) MGN + BORF | (d) MGN |
| (e) Ground Truth | (f) MGN + PIORF | (g) MGN + BORF | (h) MGN |

Figure 4: Comparison of 2D cross-sectional velocity magnitude contours for CYLINDERFLOW (a)-(d) and AIRFOIL (e)-(h) at the last time step with the largest cumulative error. It is most similar to ground truth when PIORF is applied. The closer the color is to red, the faster the velocity. The black boxes (□) highlight regions where PIORF shows particular accuracy in predicting complex flow structures. PIORF consistently achieves the closest match to ground truth on both datasets. More rollout images can be found in Appendix D.

**Setting.** We use MGN with 15 layers and maintain the same baseline rewiring methods, adjusting only dataset-specific hyperparameters. We set the velocity noise standard deviation to 0.02 in all methods. DIGL is set with `alpha` to 0.01 and `eps` at 0.4. For SDRF, we set a maximum of 10

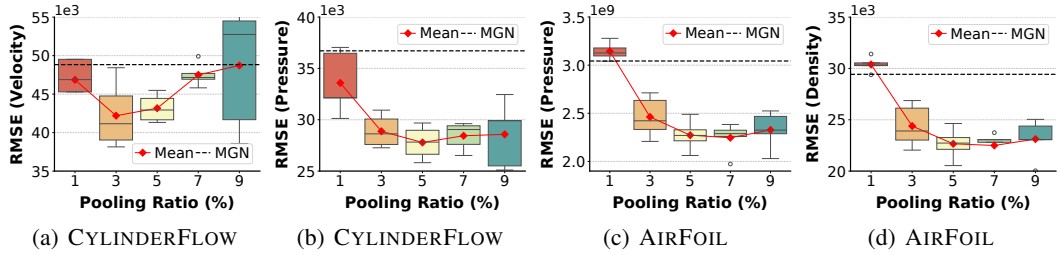

Figure 5: Sensitivity to pooling ratio $\delta$. The dashed lines represent RMSE of MGN without rewiring.

iterations and no edge removal, and for FoSR, we use an initial power of 5 and a maximum of 20 iterations. To ensure statistical significance, we repeat each experiment 5 times with different seeds.

**Results.** The timeout of BORF highlights the computational challenges in applying rewiring to the large-scale dataset. As shown in Table 3, our PIORF outperforms all baselines, achieving a 14.5% improvement in velocity RMSE over MGN. While other rewiring methods such as SDRF and FoSR show some improvements, they are significantly smaller compared to PIORF. Fig. 11 in Appendix D shows the result of the last step with different rewiring methods applied.

Table 3: Rollout-all RMSE ($\times 10^3$)

| Model | EAGLE | |
|---|---|---|
| | Velocity | Pressure |
| MGN | $2{,}280 \pm _{135}$ | $10{,}893 \pm _{632}$ |
| + DIGL | $2{,}623 \pm _{114}$ | $12{,}688 \pm _{698}$ |
| + SDRF | $2{,}186 \pm _{70}$ | $10{,}504 \pm _{297}$ |
| + FoSR | $2{,}254 \pm _{63}$ | $10{,}755 \pm _{246}$ |
| + BORF | Time-out | Time-out |
| + PIORF | $1{,}950 \pm _{28}$ | $9{,}449 \pm _{167}$ |

## 6.3 COMPUTATIONAL EFFICIENCY

Given the large scale of mesh graphs, with thousands of nodes and tens of thousands of edges (see Table 6 in Appendix B), we need to add a large number of edges to alleviate over-squashing. However, existing rewiring methods require multiple iterations to add or delete edges, leading to increased computational overhead. Fig. 6 shows the computation time required to add varying numbers of edges when rewiring one trajectory in the CYLINDERFLOW, AIRFOIL, and EAGLE datasets. PIORF maintains the lowest computation time in all datasets and edge counts. This is due to the ability of PIORF to compute all the necessary rewiring in a single pass, avoiding an iterative process. In contrast, BORF shows a steep increase in computation time as the number of added edges grows, particularly evident in EAGLE. Although SDRF and FoSR are more efficient than BORF, they still show a trend of increasing computational time, emphasizing the scalability advantage of PIORF in handling large-scale fluid dynamics simulations.

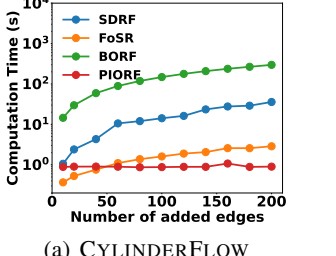 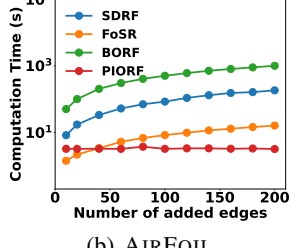 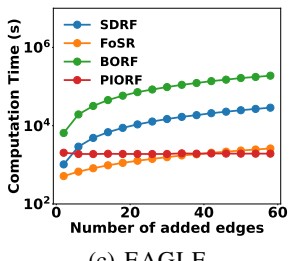

Figure 6: Comparison of computation time as the number of edges added increases.

## 6.4 ABLATION STUDIES

We conduct ablation studies to evaluate components of PIORF and Table 4 summarizes our findings.

**Choice of physical value for rewiring.** We analyze the impact of using velocity or pressure to identify nodes for edge rewiring in PIORF. For CYLINDERFLOW, an incompressible flow (Panton,

Table 4: Rollout-all RMSE ($\times 10^3$) for PIORF and the ablations.

| Ablation Model | Actions | | CYLINDERFLOW | | AIRFOIL | | |
| --- | --- | --- | --- | --- | --- | --- | --- |
| | Add | Remove | Velocity | Pressure | Velocity | Pressure | Density |
| MGN | | | $48.8 \pm 5.6$ | $36.7 \pm 2.4$ | $10,261 \pm 832$ | $3,043,186 \pm 282,514$ | $29.4 \pm 2.7$ |
| Velocity | ✓ | | $\mathbf{41.6} \pm \mathbf{3.9}$ | $\mathbf{28.9} \pm \mathbf{1.5}$ | $\mathbf{7,743} \pm \mathbf{584}$ | $\mathbf{2,245,858} \pm \mathbf{142,452}$ | $\mathbf{22.5} \pm \mathbf{1.4}$ |
| Pressure | ✓ | | $80.9 \pm 13.4$ | $75.8 \pm 15.3$ | $7,768 \pm 288$ | $2,293,481 \pm 108,098$ | $23.1 \pm 1.2$ |
| Random | ✓ | | $43.4 \pm 2.4$ | $32.3 \pm 1.2$ | $10,317 \pm 771$ | $3,115,406 \pm 230,796$ | $30.2 \pm 2.2$ |
| Only Removal | | ✓ | $42.0 \pm 2.3$ | $36.8 \pm 0.5$ | $10,890 \pm 438$ | $3,289,964 \pm 94,568$ | $31.7 \pm 0.8$ |
| Both | ✓ | ✓ | $49.0 \pm 7.5$ | $31.1 \pm 2.9$ | $7,813 \pm 551$ | $2,334,583 \pm 182,600$ | $23.4 \pm 1.9$ |
| Weighted Edges | ✓ | | $53.2 \pm 8.6$ | $44.1 \pm 4.5$ | $11,214 \pm 563$ | $3,486,655 \pm 203,277$ | $32.9 \pm 1.6$ |
| To Senders | ✓ | | $53.4 \pm 7.2$ | $35.8 \pm 1.0$ | $10,358 \pm 866$ | $3,099,548 \pm 320,553$ | $29.7 \pm 2.8$ |
| To Receivers | ✓ | | $47.9 \pm 4.9$ | $35.0 \pm 4.9$ | $10,421 \pm 704$ | $3,132,703 \pm 119,582$ | $30.3 \pm 1.2$ |

2024), velocity-based rewiring significantly outperforms pressure-based rewiring. This aligns with Bernoulli's principle for incompressible flows, where velocity changes more indicate key flow dynamics. For AIRFOIL, a compressible (Saad, 1985) and turbulent flow (Mathieu & Scott, 2000), pressure-based and velocity-based rewiring performs well and outperforms other rewiring methods.

**Effect of physical-informed node selection.** PIORF selects the nodes based on ORC-identified bottlenecks and nodes with high physical changes. To assess the impact of using physical values in this selection process, we compare our approach ("Velocity") with a modified version ("Random") where nodes are chosen based on ORC bottlenecks but the second node is selected randomly, ignoring physical values. Results show that physics-informed selection outperforms random selection.

**Effect of edge addition/removal.** We analyze the effects of edge addition ("Velocity"), removal ("Only Removal"), and both ("Both"). Removal ("Only Removal") removes the edge with the highest positive curvature. Interestingly, edge addition alone yields the best performance for all datasets, suggesting that adding new edges is beneficial than removing existing ones.

**Weighted edges.** We explore the impact of weighted edges by the L2 distance of velocity when calculating ORC in Equation (3) and Equation (4). The "Weighted Edges" results indicate that this approach does not improve performance. It means that binary edge existence might be sufficient for capturing relevant physical relationships.

**Directionality in rewiring.** We dissect the effect of directional rewiring by adding one-way edge sets. 'To Senders' is when aggregation is performed from receivers to senders, while "To Receivers" is the opposite. The results show that bidirectional rewiring outperforms unidirectional approaches.

## 7 CONCLUSIONS

We introduce PIORF as a new rewiring method that simultaneously considers the topology and physical correlation of the mesh graph and experimentally demonstrate best performance in the field of physics mesh simulation. Moreover, we show for the first time that applying our rewiring method to hierarchical GNNs and Transformer also improves model performance in mesh graph.

**Limitations and future work.** One limitation of PIORF is its dependence on the choice of physical values for rewiring. Future research could focus on developing adaptive mechanisms for selecting the most relevant physical quantities automatically. Another important direction is to extend PIORF to handle dynamic adaptive mesh refinement (Bangerth & Rannacher, 2003; Cerveny et al., 2019), which could include integrating PIORF with error estimation techniques that enable more targeted refinement in areas with large solution errors. Additionally, extending our PIORF to applications in multi-body dynamics (Choi et al., 2013), equivariant graphs (Satorras et al., 2021), and particle-based simulations (Li et al., 2018) is an important area of future work.

ETHICS STATEMENT

Our proposed PIORF is a rewiring method designed for modeling physical systems on unstructured meshes, and thus it poses no clear negative societal or ethical implications. However, potential misuse or application of the algorithm in unintended areas could result in unintended consequences.

Additionally, this paper may have implications regarding the carbon footprint and accessibility of learning algorithms. Recently, as the computational demands in machine learning research have grown, they have led to an increasing carbon footprint. Our proposed method contributes to reducing this carbon footprint by not only improving performance but also enhancing computational efficiency in such contexts.

REPRODUCIBILITY

We provide the source code for our experimental environments and the proposed method. In the future, we intend to make this source code available for the benefit of the community. PIORF source code can be found in the following: https://github.com/yuyudeep/piorf

PIORF has a single hyperparameter, the pooling ratio $\delta$. The best hyperparameter option for reproduction in each dataset is described in Section 5, along with sensitivity analysis. Additionally, the experimental settings for the proposed method and baseline can be found in Section 6.1, Section 6.2, and Appendix C.

ACKNOWLEDGEMENTS

This work was supported by the LG Display and an IITP grant funded by the Korean government (MSIT) (No. RS-2020-II201361, Artificial Intelligence Graduate School Program (Yonsei University)). K. Lee acknowledges support from the U.S. National Science Foundation under grant IIS 2338909.

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

# Supplementary Materials for "PIORF"

# Table of Contents

## A COMPARISON OF REWIRING METHODS AND COMPLEXCITY

We further compare existing graph rewiring methods with our proposed method. As shown in Table 5, our method, PIORF, takes the physical context into account, which other rewiring methods do not. This is a key characteristic of our approach, which aims to overcome the limitations of existing methods for learning fluid dynamics simulations that are not designed for this purpose.

The complexity of PIORF is $\mathcal{O}(|\mathcal{E}|d_{\max}^3)$, where $|\mathcal{E}|$ is the number of edges and $d_{\max}$ is the maximal degree. In particular, simulation datasets in fluid dynamics use thousands to tens of thousands of nodes and edges to ensure solution accuracy, so the PIORF method is advantageous in applying more edges to these datasets. One of the biggest differences is the computational cost. Existing methods such as DIGL, SDRF, FoSR, and BORF incur significant computational cost in the process of selecting which edges to rewiring to optimize their own defined objective function (See Fig. 6). Our method, on the other hand, performs the rewiring without any objective function optimization, which is beneficial in terms of computational cost. Another important difference is the number of hyperparameters. Existing rewiring methods typically require two or more hyperparameters, while our PIORF uses *only one* hyperparameter, the pooling ratio. This has the advantage of reducing the hyperparameter search space.

Table 5: Comparison of different rewiring methods and our PIORF.

| Methods | Indicator | Complexity | Geometry | Physics |
|---|---|---|---|---|
| SDRF | Balanced Forman curvature | $\mathcal{O}(|\mathcal{E}|d_{\max}^2)$ | ✓ | ✗ |
| FoSR | Spectral gap | $\mathcal{O}(\mathcal{V}^2)$ | ✓ | ✗ |
| BORF | Ollivier–Ricci curvature | $\mathcal{O}(|\mathcal{E}|d_{\max}^3)$ | ✓ | ✗ |
| PIORF | Ollivier–Ricci curvature with physical context | $\mathcal{O}(|\mathcal{E}|d_{\max}^3)$ | ✓ | ✓ |

# B  Datasets Details

CYLINDERFLOW-TINY dataset used in Fig. 1 is used to illustrate the concept of PIORF and is not desinged for evaluation. We use three public datasets for evaluation, and Table 6 shows information such as the number of cases, number of steps, number of nodes and number of edges for each dataset. AIRFOIL and EAGLE datasets are turbulent flow models, and CYLINDERFLOW is a laminar flow model. Fig. 7 shows the velocity magnitude contour image of each datasets. In all datasets, the velocity is high in areas near boundary conditions such as walls. Fig. 8 shows the distribution of ORC by each dataset. When creating a mesh, nodes with high degrees occur due to local fine mesh and boundary conditions. Red circles are nodes where the degree is 7 or higher and bottlenecks occur.

Table 6: Dataset description: Fluid dynamics behavior, number of trajectories for each data set, time step, and average number of nodes, edges, and cells in the training data set. A cell refers to an element and is a small unit that makes up a mesh. In the case of a triangular mesh, one cell consists of three nodes.

| Datasets | Behavior | Cases (Train) | Cases (Test) | Steps | Nodes (avg) | Edges (avg) | Cells (avg) |
|---|---|---|---|---|---|---|---|
| CYLINDERFLOW | Laminar | 1,000 | 100 | 600 | 1,886 | 10,848 | 3,538 |
| AIRFOIL | Turbulent | 1,000 | 100 | 600 | 5,233 | 30,898 | 10,216 |
| EAGLE | Turbulent | 947 | 118 | 990 | 3,389 | 20,023 | 6,623 |

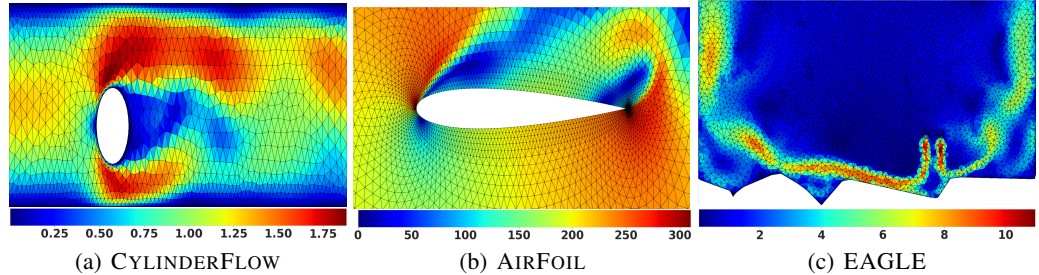

    (a) CYLINDERFLOW         (b) AIRFOIL          (c) EAGLE

Figure 7: Velocity magnitude contour image for each dataset. In all cases, changes in velocity occur in walls where fluid cannot flow.

**CYLINDERFLOW-TINY.**  CYLINDERFLOW-TINY is the dataset used to illustrate the concept of our PIORF for understanding the flow of fluid in narrow passages around a cylinder. To create CYLINDERFLOW-TINY dataset, we consider performing simulation modelling in an environment similar to that of CYLINDERFLOW. We use the Ansys Fluent® solver (Stolarski et al., 2018) to generate the dataset. The number of nodes is approximately 300 and the fluid input is air.

**CYLINDERFLOW.**  CYLINDERFLOW is important in many industrial applications, such as the cooling of cylindrical pipes, by analyzing the flow of fluid around a cylinder. The flow can exhibit laminar or turbulent flow behavior depending on factors such as flow rate, fluid density, and cylinder size. CYLINDERFLOW dataset (Pfaff et al., 2020) consists of 1,000 analysis results, with each case containing 600 time steps. The dataset contains a single cylinder, but includes a variety of Reynolds numbers, sizes, and positions.

**AIRFOIL.**  AIRFOIL is an application of aerodynamics and the most basic CFD modeling. AIRFOIL, also known as wings, is utilized in the design of airfoils and various other aerodynamic applications such as aircraft, helicopters, and spacecraft. AIRFOIL plays a central role in designing an airplane's wings to generate lift, control flight, and move through airflow. Moreover, it is very important to design an aerodynamic design that is effective in a specific range of flow conditions. AIRFOIL dataset (Pfaff et al., 2020) consists of 1,000 analysis results, with each case containing 600 time steps. The data set contains one AIRFOIL and various input conditions, such as velocity and pressure, with the fluid density changing at every step.

**EAGLE.** EAGLE is a large-scale dataset for learning non-steady fluid dynamics. This is a simulation of the airflow generated by a drone moving in a 2D environment with various boundary shapes. It is much more difficult than other datasets such as CYLINDERFLOW or AIRFOIL as it models the complex ground effect turbulence created by the drone's airflow according to its control laws. Different scene geometries produce completely different results, resulting in highly turbulent and non-periodic eddies and high flow diversity. In the field of learned simulators, EAGLE is the first to apply a dynamic mesh effect in which the shape and position of the mesh change at every time step. It accurately simulates fluid behavior by changing the drone's position over time.

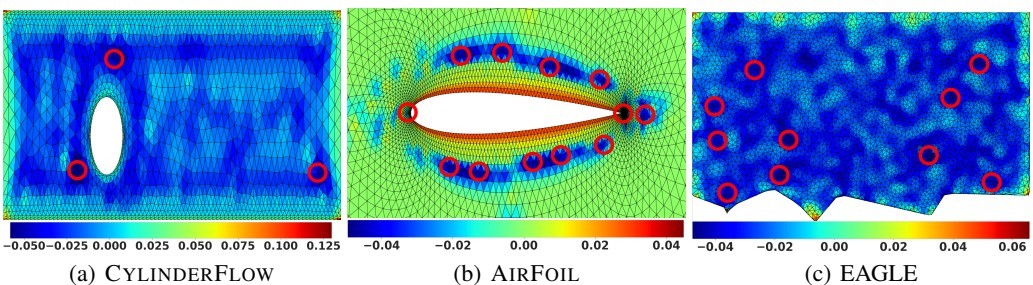

|  (a) CYLINDERFLOW  |  (b) AIRFOIL  |  (c) EAGLE  |

Figure 8: ORC distribution image for each dataset. Red circles (◯) are the nodes where the degree is high and a bottleneck occurs.

**Representative physical quantity.** The velocity refers to the speed at which a fluid moves at a specific point in space. The pressure is the force exerted by a fluid per unit area on the surfaces. The density $\rho$ is a measure of how much mass is contained within a given volume of the substance. It is defined as mass per unit volume. The density of a fluid depends on temperature and pressure. These three physical quantities are related by Bernoulli's equation. When density is constant, increasing velocity causes pressure to decrease.

## C  BASELINE DETAILS

We compare four competitive rewiring methodologies and four models. For models, MGN (Pfaff et al., 2020), BSMS, GT, and HMT are used, along with rewiring methods such as DIGL (Gasteiger et al., 2019), SDRF (Topping et al., 2021), FoSR (Karhadkar et al., 2022), and BORF (Nguyen et al., 2023). For all datasets, training steps are set to 10,000,000. Velocity noise standard deviation is 0.02 and 10 for CYLINDERFLOW and AIRFOIL datasets, respectively.

### C.1  REWIRING METHODS

For DIGL, we set `alpha` to 0.01 and use 0.4 for `eps`. For SDRF, max number of iterations is 10. Edge removal is not used. For FoSR, initial power and max number of iterations are set to 5 and 20, respectively. In the case of BORF, the max number of iterations is set to 10, and edge addition and deletion for each batch are set to 4 and 2, respectively. We use the official implementation released by the authors on GitHub for all rewiring baselines:

- DIGL: https://github.com/gasteigerjo/gdc.git
- SDRF: https://github.com/jctops/understanding-oversquashing
- FoSR: https://github.com/kedar2/FoSR
- BORF: https://github.com/hieubkvn123/revisiting-gnn-curvature

### C.2  MODELS

**MGN.** To align with the MGN methodology, we apply 15 iterations of message passing in all datasets. All MLPs have a hidden vector size of 128. Table 7 indicates the input, edge, and output features used for each dataset.

Table 7: Details of features for each dataset. $\rho_i$ is the fluid density and $\mathbf{w}_i$ is the velocity of the fluid. $\dot{\mathbf{w}}_i$ is the gradient of velocity, $\mathbf{n}_i$ is the node type, and $\mathbf{x}_i$ is the position of the node.

| Datasets | Inputs $\mathbf{m}_{ij}$ | Inputs $\mathbf{v}_i$ | Outputs $\mathbf{o}_i$ |
|---|---|---|---|
| CYLINDERFLOW | $\mathbf{x}_{ij}, \lvert\mathbf{x}_{ij}\rvert$ | $\mathbf{n}_i, \mathbf{w}_i$ | $\dot{\mathbf{w}}_i, p_i$ |
| AIRFOIL | $\mathbf{x}_{ij}, \lvert\mathbf{x}_{ij}\rvert$ | $\mathbf{n}_i, \mathbf{w}_i, \rho_i$ | $\dot{\mathbf{w}}_i, p_i, \dot{\rho}_i$ |
| EAGLE | $\mathbf{x}_{ij}, \lvert\mathbf{x}_{ij}\rvert$ | $\mathbf{n}_i, \mathbf{w}_i$ | $\dot{\mathbf{w}}_i, p_i$ |

**BSMS.** We implement the BSMS model with `Tensorflow`. And according to the best hyperparameters of BSMS, levels 7 and 9 are used for CYLINDERFLOW and AIRFOIL, respectively. Noise standard deviation is set the same as MGN. All MLPs have a hidden vector size of 128. The Encoder/decoder are set to those in MGN.

**GT.** The hidden dimension size inside its Transformer is set to 128. FFN used three linear layers and two ReLU activations. To ensure numerical stability, the results obtained with the exponential term within the softmax function are constrained to fall in the range of $[-2, 2]$. We use the FFN without using positional encoding. There are 15 transformer blocks with 4 heads. The encoder/decoder are set to those in MGN.

**HMT.** Because contact edges are not used in fluid datasets, we only use HMT among the modules of the HCMT model. The hidden dimension size of HMT is set to 128, and both FFN and numerical stability are set to the same as GT. There are 15 transformer blocks with 4 heads. The encoder/decoder are set to those in MGN.

We use the official implementation released by the authors on GitHub for all baselines models:

- MGN: https://github.com/google-deepmind/deepmind-research/tree/master/meshgraphnets
- BSMS: https://github.com/Eydcao/BSMS-GNN
- GT: https://github.com/graphdeeplearning/graphtransformer
- HMT: https://github.com/yuyudeep/hcmt

# D OTHER VARIABLE CONTOUR AND ROLLOUT FIGURES

Figs. 9 to 11 are rollout images of CYLINDERFLOW, AIRFOIL, and EAGLE, from the last time step with the highest cumulative error.

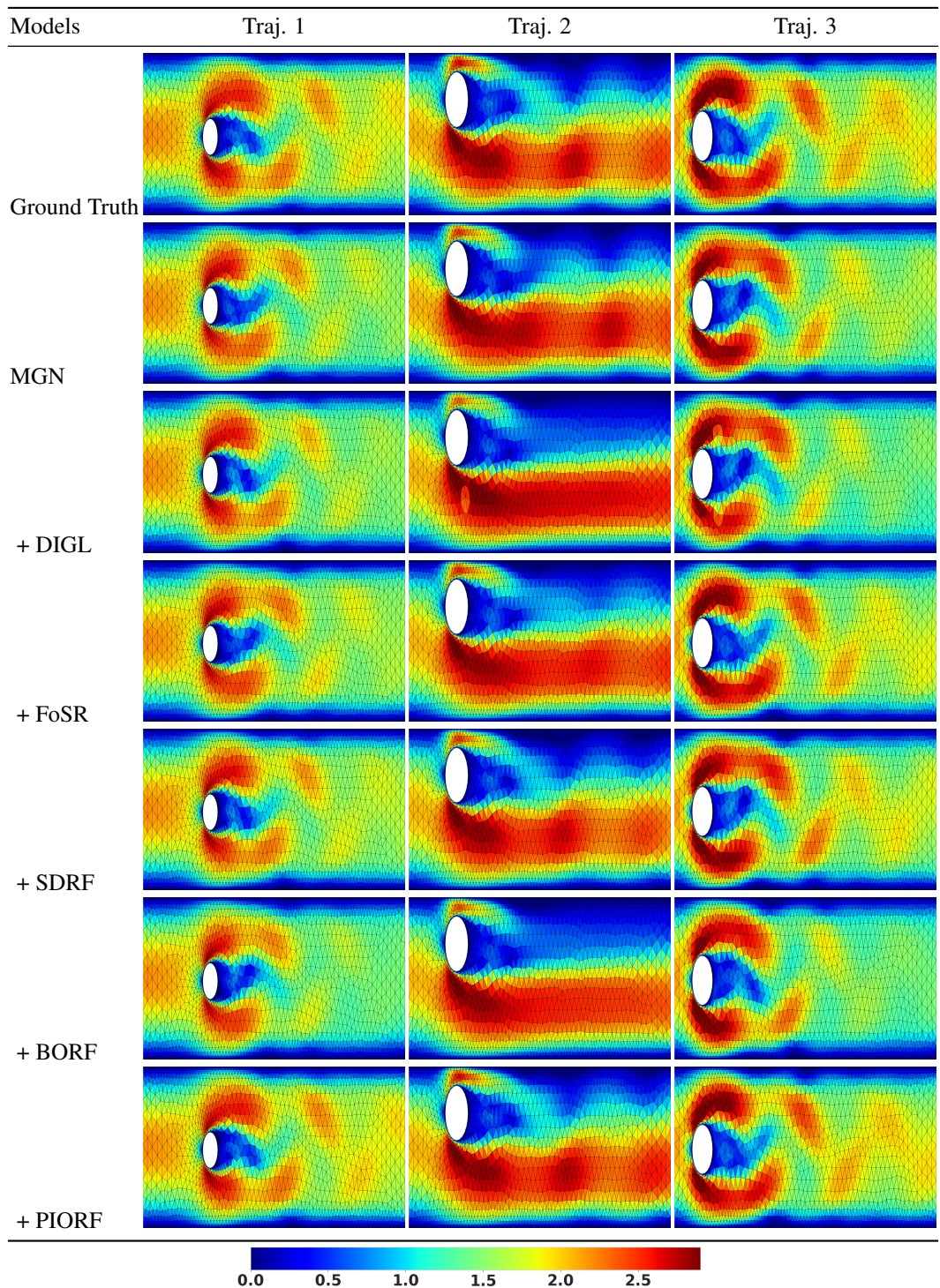

Figure 9: The velocity magnitude contours of various rewiring methods compared to the ground truth at CYLINDERFLOW

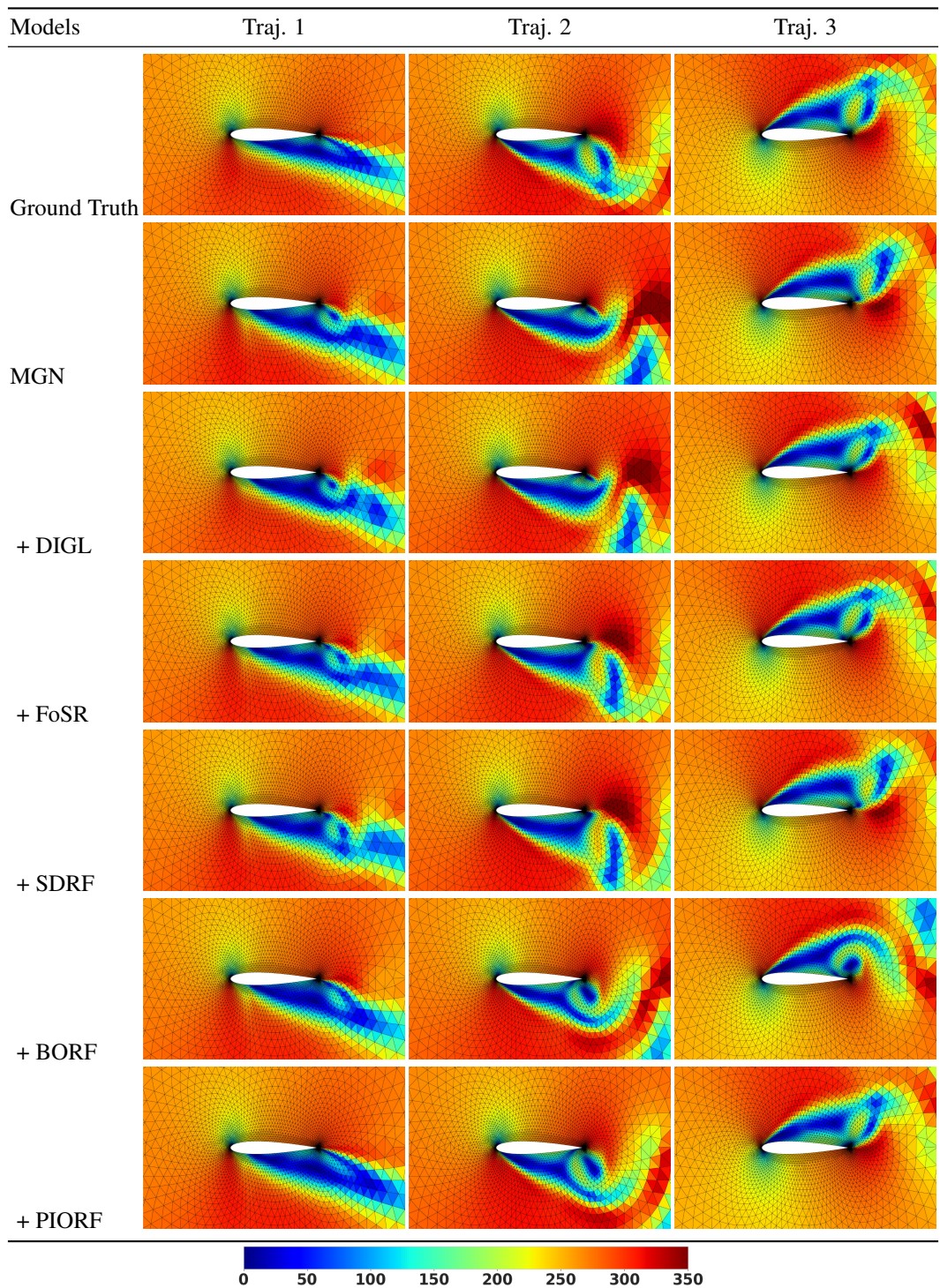

Figure 10: The velocity magnitude contours of various rewiring methods compared to the ground truth at AIRFOIL

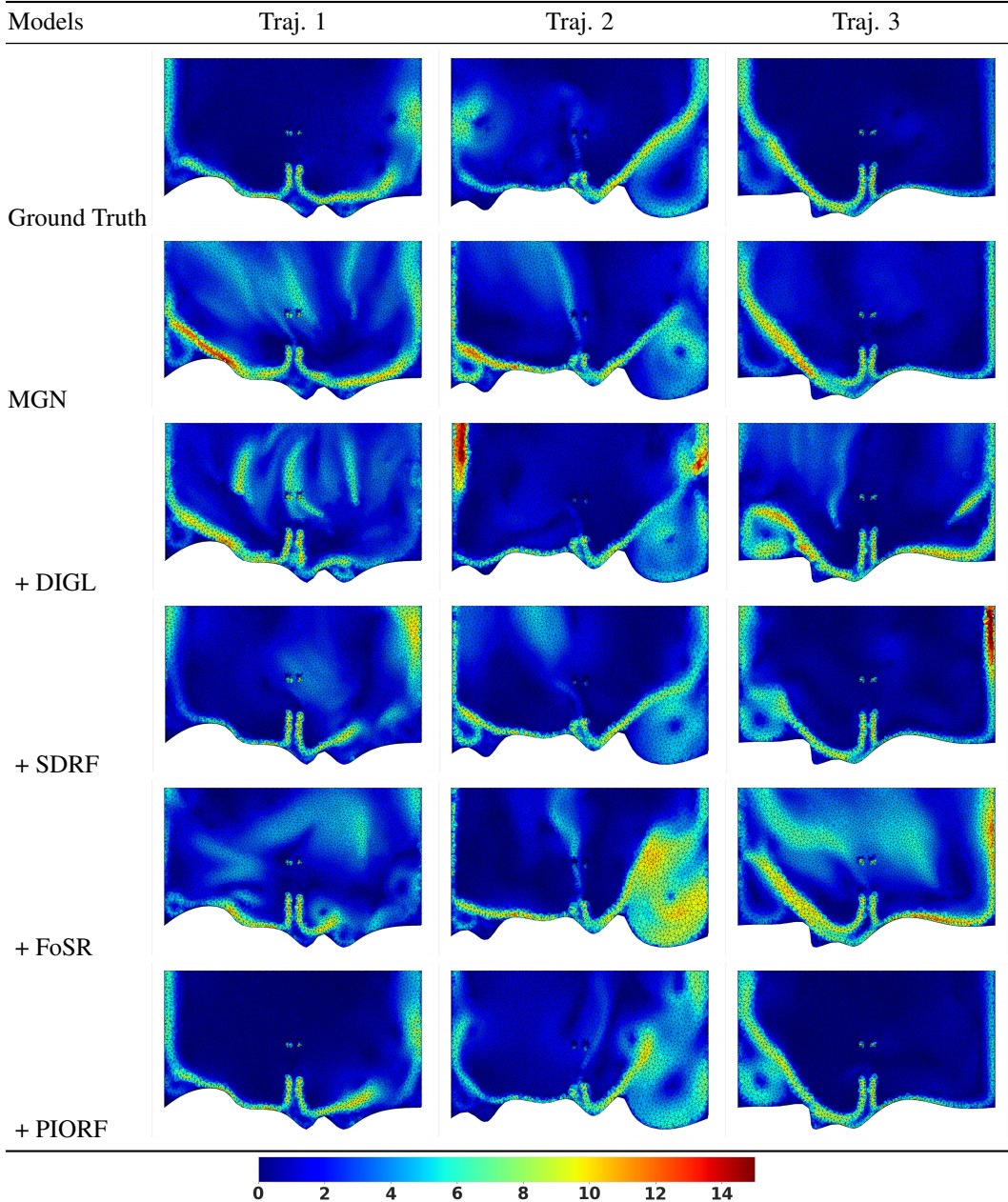

Figure 11: The velocity magnitude contours of various rewiring methods compared to the ground truth at EAGLE.

# E  NOTATIONS

Table 8 outlines the key notations used in the paper.

Table 8: Notation summary.

| Name | index |
|---|---|
| senders | $i$ |
| receivers | $j$ |
| ORC(edge) | $\kappa(i,j)$ |
| ORC(node) | $\gamma_i$ |
| nodes | $u, v$ |
| the shortest distance | $d(u,v)$ |
| distribution of 1-step random walk from node $u$ | $\mathbf{m}_u$ |
| L1 Wassertein transport distance | $W_1(\mu_i, \mu_j)$ |
| sets of nodes | $\mathcal{V}$ |
| sets of edges | $\mathcal{E}$ |
| ORC Pooling Ratio | $\delta$ |
| Inputs(edge features) | $\mathbf{m}_{ij}$ |
| Inputs(node features) | $\mathbf{v}_i$ |
| Outputs | $\mathbf{o}_i$ |
| Edge Hidden Features | $\mathbf{e}_{ij}$ |
| Updated Edge Hidden Features | $\mathbf{e}'_{ij}$ |
| Node Hidden Features | $\mathbf{v}_{ij}$ |
| Updated Node Hidden Features | $\mathbf{v}'_{ij}$ |
| Node MLP | $\mathbf{f}^V$ |
| Edge MLP | $\mathbf{f}^E$ |
| Number of nodes | $|\mathcal{V}|$ |
| Number of edges | $|\mathcal{E}|$ |
| Mesh Positions | $\mathbf{x}_i$ |
| Relative Mesh Positions | $\mathbf{x}_{ij}$ |
| Norm Relative Mesh Positions | $|\mathbf{x}_{ij}|$ |
| Node Type | $\mathbf{n}_i$ |
| Velocity | $\mathbf{w}_i$ |
| Velocity Gradient | $\dot{\mathbf{w}}_i$ |
| Predicted Velocity Gradient | $\hat{\dot{\mathbf{w}}}_i$ |
| Pressure | $p_i$ |
| Predicted Pressure | $\hat{p}_i$ |
| Density | $\rho_i$ |
| Density Gradient | $\dot{\rho}_i$ |
| Predicted Density Gradient | $\hat{\dot{\rho}}_i$ |

## F    ADDITIONAL ABLATION STUDIES

Our proposed rewiring method has node selection steps that depend on ingredients such as degree, ORC, and physical context. We conduct additional ablation studies to evaluate performance across different ingredient selections. The pooling ratio for all experiments is 3%.

Table 9 shows performance based on ingredient selection. The first step is to select nodes based on curvature("Former", Algorithm 1 lines 3-4), and the second is to select nodes based on physical context("Latter", Algorithm 1 lines 5-6). We define the following four rewiring methods for ablation studies: i) "Ablation 1", where the former refers to high degree and the latter to physics, ii) "Ablation 2", where the former refers to random and the latter to physics, iii) "Ablation 3", where the former refers to random and the latter to random, and iv) "Ablation 4", where the former refers to ORC and the latter to random.

In CylinderFlow, "Ablation 1" shows results with high-degree selection, achieving improved performance compared to MGN. However, it underperforms relative to PIORF, as it exhibits varying curvature values for the same degree. "Ablation 2" and "Ablation 4" show performance based on the choice of physical context and ORC, respectively. Both outperform MGN, and Physical Context provides slightly better performance than ORC. "Ablation 3" is the result of randomly selecting both the former and the latter and adding edges, and is similar to the performance of MGN.

Table 9: Rollout-all RMSE ($\times 10^3$) for PIORF and the ablations.

| Model | Ingredient | | CYLINDERFLOW | | AIRFOIL | | |
|---|---|---|---|---|---|---|---|
| | Former | Latter | Velocity | Pressure | Velocity | Pressure | Density |
| MGN | | | $48.8 \pm 5.6$ | $36.7 \pm 2.4$ | $10,261 \pm 832$ | $3,043,186 \pm 282,514$ | $29.4 \pm 2.7$ |
| PIORF | ORC | Physics | $41.6 \pm 3.9$ | $28.9 \pm 1.5$ | $7,743 \pm 584$ | $2,245,858 \pm 142,452$ | $22.5 \pm 1.4$ |
| Ablation 1 | Degree | Physics | $44.9 \pm 5.7$ | $33.3 \pm 1.1$ | $10,379 \pm 607$ | $3,065,807 \pm 276,373$ | $29.6 \pm 2.6$ |
| Ablation 2 | Random | Physics | $44.6 \pm 0.8$ | $31.1 \pm 1.0$ | $10,150 \pm 505$ | $2,936,397 \pm 177,037$ | $28.4 \pm 1.8$ |
| Ablation 3 | Random | Random | $48.4 \pm 1.6$ | $35.3 \pm 0.9$ | $11,220 \pm 538$ | $3,305,957 \pm 176,642$ | $31.9 \pm 1.7$ |
| Ablation 4 | ORC | Random | $43.4 \pm 2.4$ | $32.3 \pm 1.2$ | $10,317 \pm 771$ | $3,115,406 \pm 230,796$ | $30.2 \pm 2.2$ |

## G    ADDITIONAL DISCUSSION

### G.1    GRAPH TOPOLOGY CHANGES.

We analyze changes in graph topology in each dataset. Fig. 12 shows a comparison of curvature distributions between the original graph and the graph using PIORF. The graph constructed after applying PIORF shows the removal of highly negative curvatures that cause bottlenecks (Topping et al., 2021).

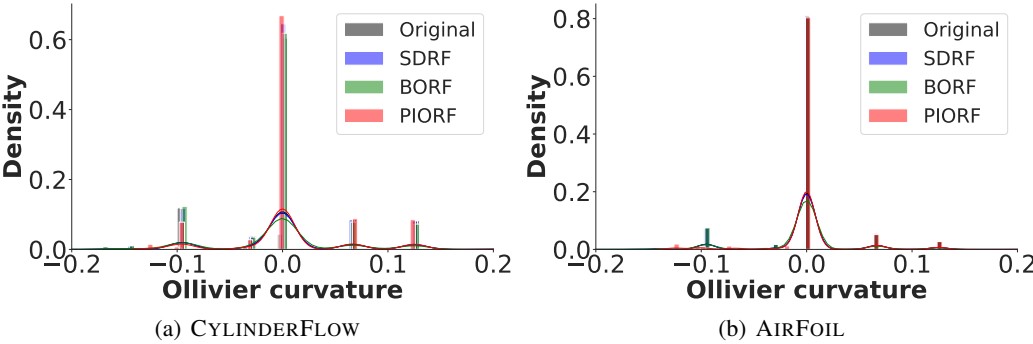

(a) CYLINDERFLOW                    (b) AIRFOIL

Figure 12: Comparison of curvature distributions between the original graph and the graph using PIORF. The $x$-axis represents the Ollivier curvature of the edges, and the plots show a kernel density estimate of the curvature distribution.

## G.2 EFFECTIVE RESISTANCE ON GRAPHS.

The effective resistance (Black et al., 2023) provides a metric for measuring over-squashing. We randomly pick up 10,000 sample graphs from each dataset and analyze the total resistance (the sum of the effective resistance between all pairs of nodes). Table 10 shows the total effective resistance results in the original graph and the graph after applying the PIORF method in each dataset. The total effective resistance is significantly reduced, which indicates that the bottleneck is alleviated and enables long-range propagation.

Table 10: Total resistance for our PIORF and baselines.

| Methods | CylinderFlow | Airfoil |
|---------|--------------|---------|
| MGN | 2,491,084 | 15,644,891 |
| SDRF | 2,487,198 | 15,628,620 |
| BORF | 2,398,661 | 15,403,149 |
| PIORF | 1,653,709 | 10,140,834 |

## G.3 RELATIONSHIP BETWEEN ACCUMULATED ERROR AND VELOCITY GRADIENT.

In the field of dynamics learning simulations, such as MGN, the model iteratively predicts the next step. The longer the simulation steps, the more accumulated error occurs during inference. Fig. 13 shows the change in accumulated error and the gradient of velocity for each step after applying PIORF. Areas with significant accumulated errors depend on the velocity gradient. In PIORF, which reflects this physical quantity, the overall accumulated error is reduced compared to the original.

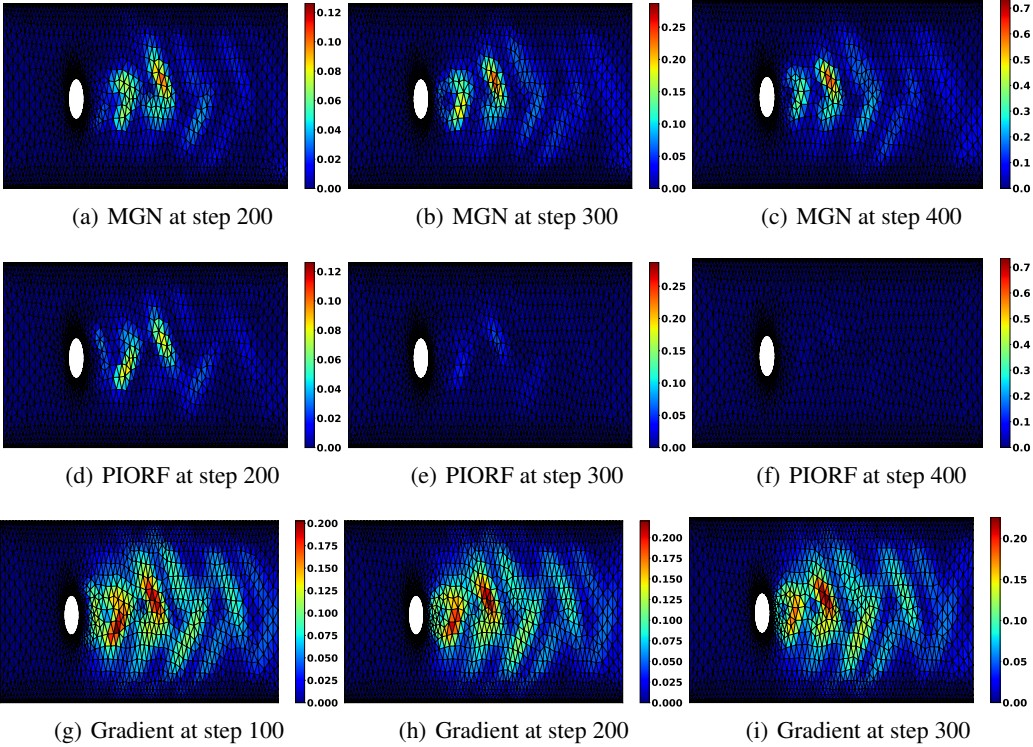

Figure 13: Comparison of accumulated error distribution and gradient magnitude of velocity distribution according to PIORF application.

## G.4 DISCUSSION ON ORC IN PIORF

In studies proposing rewiring methods in the field of graph machine learning, metrics with various curvature concepts are typically used to optimize edge addition or removal by maximizing their values (Topping et al., 2021; Nguyen et al., 2023). In physics simulation, alternatives to our ORC-based approach could include methods such as SDRF (Topping et al., 2021) and BORF (Nguyen et al., 2023) that use different metrics for optimization. For example, SDRF uses a balanced Forman curvature, which provides a more conservative estimate compared to ORC (Nguyen et al., 2023, Lemma 4.1). While BORF similarly uses ORC for rewiring, our experiments in Table 1 demonstrate why it is less suitable for physics simulation.

Metrics that can be presented as alternatives with a similar role to ORC are Forman curvature (Sreejith et al., 2016) and betweenness centrality (Barthelemy, 2004). However, these metrics do not capture the area near the boundary conditions effectively. This region is where fluid flow changes and is also crucial from a domain knowledge. While Forman curvature, based on the graph Laplacian, is easier and faster to compute than Ollivier-Ricci curvature, it is less geometric (Ni et al., 2019). We choose ORC specifically because it better captures geometric characteristic, particularly around boundary conditions where fluid flow changes dramatically. Betweenness centrality could be used for source node selection, its high complexity $\mathcal{O}(|\mathcal{V}||\mathcal{E}|)$ and need for global information make it impractical for mesh graphs with thousands of nodes and edges.

