# OpenReview forum: "PIORF: Physics-Informed Ollivier-Ricci Flow for Long–Range Interactions in Mesh Graph Neural Networks"
_ICLR.cc/2025/Conference — ICLR 2025 Poster_

### Official Review · Reviewer_SBAf · 2024-10-29

**Soundness:** 2
**Presentation:** 3
**Contribution:** 2
**Rating:** 5
**Confidence:** 4

**Summary:**

The paper introduces a method called PIORF that enhances long-range interactions in mesh graph neural networks for fluid dynamics simulations. It uses Ollivier–Ricci curvature (ORC) to identify bottleneck regions in the graph and connects these areas with nodes having high velocity gradients. The authors evaluate PIORF on three fluid dynamics datasets and choose multiple backbone GNNs to demonstrate its effectiveness. The results show that PIORF outperforms existing graph rewiring methods in fluid simulation.

**Strengths:**

1. The paper studies the over-squashing problem in fluid simulation.
2. The proposed methods consider both the graph topology and physical quantity.
3. The authors conduct experiments on various backbones.

**Weaknesses:**

I appreciate the authors' effort in exploring the over-squashing problem in fluid dynamic learning. However, multiple issues are still underexplored from my point of view. The details are as follows:

1. Most graph rewriting methods study the classification problem in graphs like citation networks and social networks. The significance of over-squashing in fluid simulation is underexplored.  Whether and how does it affect learning performance? One observation that raises such a question is that in Table 1, existing graph rewriting methods enhance errors in most cases, indicating that over-squashing might not be the key issue in learning fluid dynamics.

2. For the method, connecting the nodes with the highest velocity gradient is straightforward. How does the author avoid connecting nodes with low influence but high-velocity gradients?

3. What is the ratio of negative ORC? Do you consider it to determine the pooling ratio?

4. The authors do not provide time complexity.

5. From Table 4, Random and Only Removal also significantly reduces model errors. Do they alleviate the over-squashing problem?

**Questions:**

Overall, my concerns are as follows:

1. Whether over-squashing exist in or significantly affect fluid dynamic learning?
2. How does PIORF alleviate over-squashing? more insights into model designs.
3. Does the performance improvement result from that PIORF alleviate over-squashing?

**Details Of Ethics Concerns:**

N/A.

---

> ### Author Response · Authors · 2024-11-22
>
> We appreciate Reviewer SBAf for the positive feedback and insightful comment, highlighting our strengths in
> * The paper first  studies the over-squashing problem in fluid simulation.
> * The proposed methods consider both the graph topology and physical quantity
>
> Below, we carefully address your concerns. We uploaded the revised paper, where changes are highlighted in red.
>
> **W1. Most graph rewriting methods study the classification problem in graphs like citation networks and social networks. The significance of over-squashing in fluid simulation is underexplored. Whether and how does it affect learning performance?. One observation that raises such a question is that in Table 1, existing graph rewriting methods enhance errors in most cases, indicating that over-squashing might not be the key issue in learning fluid dynamics**
>
> In the field of dynamics learning simulations, such as MGN, the model iteratively predicts the next step. Since a single trajectory consists of multiple steps, existing baseline methods, which form a single graph for the entire trajectory, are not suitable for this field. In contrast, in PIORF, physical quantities change at each step within a single trajectory, resulting in a different graph being constructed each time.
>
> **W2. For the method, connecting the nodes with the highest velocity gradient is straightforward. How does the author avoid connecting nodes with low influence but high-velocity gradients?**
>
> In fluid mechanics, governing equations such as Navier Stokes are helpful in predicting fluid flow because they depend on gradients of velocity[1]. Therefore, high-velocity gradients are an important element in dynamics and have an equally important influence on learning. We confirmed that the accumulated error is related to the velocity gradient, and the result of discussion is added to Appendix G.3.
>
> >[1] Charles Meneveau. "Lagrangian dynamics and models of the velocity gradient tensor in turbulent flows." Annual Review of Fluid Mechanics, 43(1):219–245, 2011.
>
>
>
> **W3. What is the ratio of negative ORC? Do you consider it to determine the pooling ratio?**
>
> The ratio of negative ORC (<0) for CylinderFlow and Airfoil are 21.88% and 14.71%, respectively. We do not consider the ratio of negative ORC. The pooling ratio is the only hyperparameter of PIORF and is adjusted to find the optimal value. The ORC distribution for each dataset is added to Appendix G.1.
>
>
> **W4. The authors do not provide time complexity.**
>
> The complexity of PIORF is $\mathcal{O}(|\mathcal{E}|d^3_{\mathrm{max}})$, where $|\mathcal{E}|$ is the number of edges and $d_\mathrm{max}$ is the maximal degree. The complexity is added to Appendix A and highlighted in red.
>
>
>
> **W5. From Table 4, Random and Only Removal also significantly reduces model errors. Do they alleviate the over-squashing problem?**
>
> In the case of "Random", this ablation study evaluates the influence of physical context. While shortcut edges are added dynamically at each step, which helps alleviate over-squashing from a topological perspective, the results suggest that physical phenomena must also be considered, as evidenced by the comparison with PIORF.
>
> In the case of "Remove Only", edges with high positive curvature are removed, which does not indicate over-squashing problems. Additionally, except for the velocity RMSE in CylinderFlow, all other errors increase compared to MGN. The experimental conditions for "Only Removal" are missing, so they are added and marked in red.

---

> ### Author Response · Authors · 2024-11-22
>
> **Q1. Whether over-squashing exist in or significantly affect fluid dynamic learning?**
>
> Edges with high negative curvature[1] and high degree nodes[2] cause graph bottlenecks and over-squshing. The mesh graph also contains negative curvature, which can potentially lead to over-squashing. The change in graph topology before and after applying PIORF shows that high negative curvature is removed and the overall ratio of negative curvature is reduced. Furthermore, existing rewiring methods do not remove this negative curvature, indicating that they are not suitable for mesh graphs. The results of graph topology changes are added to Appendix G.1
>
> In fluid dynamics, which primarily uses triangular meshes, the regions excluding those near the boundary conditions are composed of hexagonal meshes. However, in boundary condition areas such as walls, nodes with high degrees and edges with negative curvature are created, which can lead to potential bottlenecks. In addition, since the nodes around these boundary conditions are areas where the fluid flow changes, efficient message passing to these nodes is crucial, as they are highly important from both a learning and domain knowledge perspective.
>
> > [1] Jake Topping, et al. "Understanding over-squashing and bottlenecks on graphs via curvature". ICLR, 2022.
> > [2] Andreea Deac, et al. "Expander graph propagation". ICML, 2022.
>
>
> **Q2. How does PIORF alleviate over-squashing? more insights into model designs.**
>
> We design the model considering the following points.
>
> * Learning dynamics models generate the accumulated error through iterative predictions. Existing methods are ineffective because only a single graph is constructed within each trajectory.
> * The area around the boundary conditions can cause potential over-squashing.
>
> In fluid dynamics, the gradient of velocity is an important physical quantity and is equally crucial in the learning process. We empirically find that the gradient of the velocity is related to the accumulated error (Please refer to Appdendix G.3). Therefore, by considering the physical context and adding edges with boundary condition nodes, over-squashing can be alleviated and even long-range interaction effects can be considered.
>
>
> **Q3. Does the performance improvement result from that PIORF alleviate over-squashing**
>
> In the added discussion (Appdendix G), we explain the alleviation of over-squashing through changes in graph topology and total resistance after applying PIORF. The mesh graph suggests that a rewiring method which considers physical context is more effective in several models, such as MGN, BSMS, GT, and HMT.

---

> > ### Comment · Reviewer_SBAf · 2024-11-26
> >
> > Thanks for your rebuttal and additional experiments. Some of my concerns are addressed. However, the following issues still remain:
> >
> > * For Appendix G1, how do you sample graphs? Are 100 graphs representative of CylinderFlow and Airfoil which has over 600, 000 graphs?
> > * The response to W1 further reminds me that the author does not explain how you adapt existing graph rewriting methods to multi-step fluid dynamics simulation.
> > * From the results of Appendix G1, it seems the distribution is different from the citation networks and social networks (their ORC mean and std can be found in the Appendix of BORF, which is higher). Thus, I am still wondering about the significance of applying graph rewriting methods in fluid dynamic learning.
> > * Existing rewiring methods (e.g., BORF) also remove edges with minimal curvature (generally negative).
> >
> > Therefore, I will maintain my score.

---

> > > ### Author Response · Authors · 2024-11-27
> > >
> > > We are grateful for your assistance and constructive feedback. Your insights have been invaluable in improving the quality. We greatly appreciate your valuable insights into the overs-quashing problem in fluid dyanmics.
> > >
> > > **Q1. For Appendix G1, how do you sample graphs? Are 100 graphs representative of CylinderFlow and Airfoil which has over 600,000 graphs?**
> > >
> > > We apologize for analyzing only 100 samples due to lack of time. We confirmed that the results showed no significant changes after 100. We re-uploaded the analysis results for the entire data.
> > >
> > > **Q2. The response to W1 further reminds me that the author does not explain how you adapt existing graph rewriting methods to multi-step fluid dynamics simulation.**
> > >
> > > This indicates that existing graph rewriting methods cannot effectively alleviate over-squashing that occurs in multi-step dynamics. These methods overlook such factors and apply the same graph to all steps within a trajectory from a topological perspective. Moreover, rewiring methods can be applied through preprocessing, regardless of the model. Additionally, the results in Table 1 show that SDRF and FoSR outperform MGN in CylinderFlow, while BORF outperforms MGN in Pressure and Density in AirFoil.
> > >
> > >
> > > **Q3. From the results of Appendix G1, it seems the distribution is different from the citation networks and social networks (their ORC mean and std can be found in the Appendix of BORF, which is higher). Thus, I am still wondering about the significance of applying graph rewriting methods in fluid dynamic learning.**
> > >
> > > Since the dataset used in citation networks and social networks primarily deals with classification problems, it exhibits an ORC distribution that differs from that of dynamic mesh graphs used to solve dynamic problems. However, because negative curvature exists in mesh graphs, it can lead to potential over-squashing problems, and multi-stepping causes accumulated errors, which accelerates the over-squashing issue in dynamics. We believe that these multi-level features require us to take the physical context into account.
> > >
> > >
> > > **Q4. Existing rewiring methods (e.g., BORF) also remove edges with minimal curvature (generally negative).**
> > >
> > > To the best of our knowledge, according to [BORF Algorithm 1](https://arxiv.org/pdf/2211.15779), edges for addition (lines 4-7) find minimal curvature, and edges for removal (lines 8-9) find maximal curvature.
> > >
> > > We understand that this may be inconvenient, but could you kindly review it again.

---

> ### Author Response · Authors · 2024-11-24
>
> Dear Reviewer SBAF,
>
> We appreciate your comments on helping us improve our paper in many aspects. Since we have only a few days until the discussion period ends, we wanted to remind you of our responses to your valuable feedback.
> We hope that we have clarified your concerns and questions. If we have satisfactorily addressed your concerns and questions, we kindly ask you to reconsider your score or let us know if you have any follow-up questions.
>
> Best,
>
> Authors

---

> ### Author Response · Authors · 2024-12-01
>
> Dear Reviewer SBAf,
>
> We are writing to follow up on our recent responses and remind you that we have just a few days remaining until the discussion period concludes on **December 2nd**.
>
> We would like to restate that we have made comprehensive efforts in our responses to address your remaining concerns. We believe our responses address your concerns regarding our method and its significance in fluid mechanics applications.
>
> ***If you have had the opportunity to review these responses, we greatly appreciate your consideration in potentially revising your score.*** We remain open to further discussion if you have any remaining questions.
>
> Thank you for your ongoing engagement and considerate review process.
>
> Best regards,
>
> Authors

---

> > ### Comment · Reviewer_SBAf · 2024-12-02
> >
> > Dear Authors,
> >
> > Thanks for your additional responses. I have raised my score to 5. However, I believe there is still a need to strengthen the necessity of addressing the over-squashing in physical dynamic learning and provide deeper insights into how PIORF addresses the over-squashing problem.

---

> > > ### Author Response · Authors · 2024-12-03
> > >
> > > We truly appreciate that you raise the score, but we would also like to understand if there are any remaining concerns preventing you from giving us an below acceptance score. We also appreciate the opportunity to address the reviewer’s remaining concern regarding the issue of over-squashing in physical dynamic learning.
> > >
> > > We would like to further address the reviewer’s remaining concerns by providing additional explanations in the context of: 1) "necessity of addressing the over-squashing in physical dynamic learning", 2) "deeper insight into how PIORF solves the problem of over-squashing.", and 3) "additional ablation study".
> > >
> > > Although the remaining rebuttal period is limited, we are doing our best to provide a thorough discussion.
> > >
> > > **1:** In general, the mesh structure shows a planar geometry similar to that of grid networks in most regions. However, near boundary conditions and in areas with mesh refinement, the curvature properties can vary significantly. Our empirical analysis indicates that nodes with high degrees can introduce areas of negative curvature and partially show characteristics of tree-like hyperbolic geometry. This means our mesh graphs are hybrid structures - mostly planar but with localized regions that can show hyperbolic characteristics, particularly around high-degree nodes created by mesh refinement or boundary conditions. By analyzing these features in the mesh, as shown in Figure 3a in Section 5, we empirically confirmed that the mesh graph used in the field of fluid dynamics exhibits the creation of negative curvature around high-degree nodes. These findings emphasize the relationship between the mesh configuration, boundary conditions, and the risk of information bottlenecks in GNNs used for fluid dynamics simulations.
> > >
> > > **2:** In fluid dynamics, governing equations such as Navier Stokes are helpful in predicting fluid flow because they depend on gradients of velocity. the distinction between laminar and turbulent flows, as quantified by velocity and the Reynolds numbers, is important for understanding system behavior. Velocity gradients are an important element in dynamics and have an equally important influence on learning. We confirmed that the accumulated error is related to the velocity gradient, and the result of discussion is added to Appendix G.3. PIORF effectively integrates this physical context into graph rewiring by adding edges between nodes with significant velocity differences. This allows the model to help with long-range interactions to better simulate real-world phenomena such as fluid turbulence. Furthermore, We provided a theory-based intuitive explanation of how PIORF's appropriate edge addition can mitigate over-squashing from an effective resistance perspective. Please refer to Reviewer 5Uy1 [Q2.](https://openreview.net/forum?id=qkBBHixPow&noteId=ZEoZimtev8)
> > >
> > > **3:** Our proposed rewiring method includes node selection steps that depend on factors such as degree, ORC, and physical context. To evaluate performance across different factor selections, we conducted additional ablation studies. Table shows performance based on ingredient selection in CylinderFlow. The first step is to select nodes based on curvature("Former", Algorithm 1 lines 3-4), and the second is to select nodes based on physical context("Latter", Algorithm 1 lines 5-6). The results of additional experiments are added to Appendix F.
> > > |   Method   | Former |      Latter      | Velocity RMSE | Pressure RMSE |
> > > |:----------:|:------:|:----------------:|:-------------:|:-------------:|
> > > |  MGN       |        |                  |      48.8     |      36.7     |
> > > |    PIORF   |   ORC  | Physical context |      41.6     |      28.9     |
> > > | Ablation 1 | Degree | Physical context |      44.9     |      33.3     |
> > > | Ablation 2 | Random | Physical context |      44.6     |      31.1     |
> > > | Ablation 3 | Random |      Random      |      48.4     |      35.3     |
> > > | Ablation 4 |   ORC  |      Random      |      43.4     |      32.3     |

---

### Official Review · Reviewer_isvf · 2024-11-02

**Soundness:** 3
**Presentation:** 4
**Contribution:** 3
**Rating:** 6
**Confidence:** 4

**Summary:**

This paper introduces an enhancement in mesh-based graph network simulators by addressing the over-squashing problem using a novel physics-informed rewiring approach. Notably, the method is able to scale to large scale fluid simulations due to the controlled complexity in performing rewiring. Experimentally, the proposed approach achieves notable improvement over existing mesh-based graph networks across three benchmarks, showcasing its efficacy.

**Strengths:**

1. The paper is well written and the proposed approach is easy to follow.

2. The motivation is very clear with an elegant solution based on ORC.

3. The discussions on the improved efficiency of the proposed approach over existing rewiring methods should be appreciated.

4. Experiments are thorough and the results are well presented.

**Weaknesses:**

These are some minor weaknesses which do not substantially hurt the paper but would be good to add more discussions.

1. The proposed approach is mainly developed upon mesh-based representations. Adding more discussions to whether this could be generalized to ubiquitous graph network simulators (e.g., with particle representations or rigid bodies) would enhance the contribution.

2. Some metrics are not clear to readers, e.g., Pressure, Density. It would be helpful to provide some explanations on the actual physical interpretation of these quantities, in the context of the considered simulation environments.

3. All datasets seem to be artificial and adding more experiments (if applicable) on some real-world datasets would significantly enhance the paper.

**Questions:**

Q1. Is the approach also applicable to particle-based simulations? e.g., [1].

Q2. Could the approach be combined with other physical inductive biases like equivariance [2], [3], [4]? It would be interesting to have some of these discussions in the paper or as related literature.

Q3. Could the approach be applied together with constraint-aware graph simulators like [5]?

Q4. Are there any real-world dataset on which the model can be evaluated? Practitioners would be more exciting to see how the method performs in actual real-world scenarios.

[1] Li et al. Learning particle dynamics for manipulating rigid bodies, deformable objects, and fluids.

[2] Satorras et al. E(n)-equivariant Graph Neural Networks.

[3] Huang et al. Equivariant Graph Mechanics Networks with Constraints.

[4] Han et al. Learning Physical Dynamics with Subequivariant Graph Neural Networks.

[5] Rubanova et al. Constraint-based graph network simulator.

---

> ### Author Response · Authors · 2024-11-22
>
> We appreciate Reviewer isvf for the positive feedback and insightful comment, highlighting our strengths in
> * The paper is well written and the proposed approach is easy to follow
> * The motivation is very clear with an elegant solution based on ORC
>
> Below, we carefully address your concerns. We uploaded the revised paper, where changes are highlighted in red.
>
> **W1. The proposed approach is mainly developed upon mesh-based representations. Adding more discussions to whether this could be generalized to ubiquitous graph network simulators (e.g., with particle representations or rigid bodies) would enhance the contribution.**
>
> Thanks for the nice suggestion. Although we have looked into only the fluid dynamics problems in this paper, as pointed out by the reviewer, we expect our approach can be further extended into other types of physical processes. As shown in [MGN](https://arxiv.org/abs/2010.03409) paper, there are diverse sets of physical processes that can be simulated over mesh-graph-net (MGN) framework (including rigid/soft-body dyanmics) and these simulations could be benefited by the proposed approach in terms of improved computational efficiency.
>
>
> **W2. Some metrics are not clear to readers, e.g., Pressure, Density. It would be helpful to provide some explanations on the actual physical interpretation of these quantities, in the context of the considered simulation environments.**
>
> The Navies Stokes equation, which is the governing equation in fluid mechanics, depends on physical quantities such as velocity, pressure, and density. The velocity refers to the speed at which a fluid moves at a specific point in space. The pressure is the force exerted by a fluid per unit area on the surfaces. The density is a measure of how much mass is contained within a given volume of the substance. Additional descriptions of physical quantities are added in Appendix B.
>
> **W3. All datasets seem to be artificial and adding more experiments (if applicable) on some real-world datasets would significantly enhance the paper.**
>
> This is a good suggestion, however, Unfortunately, we do not have access to real-world datasets. In industry, such datasets are typically not made public due to security concerns. We also want to evaluate how effective PIORF will be on real-world datasets. However, we attempted to include well-established dataset into our benchmark problem (for example, large-scale benchmark dataset in EAGLE, which consists of 270 GB of Navier--Stokes equations solution snapshots).
>
> **Q1. Is the approach also applicable to particle-based simulations?**
>
> We expect that PIORF can be applied to particle-based simulations, although some technical or implementational modifications may be required. In particle-based simulations, the distance between nodes is calculated, and edges are created if the nodes are within a predefined radius (a hyperparameter). Therefore, PIORF can be applied considering curvatures and physical contexts.
>
> **Q2. Could the approach be combined with other physical inductive biases like equivariance? It would be interesting to have some of these discussions in the paper or as related literature.**
>
> Thanks for the nice suggestion. The advantage of rewiring is that it can be implemented by constructing a new graph through prior data processing without modifying the model itself. Therefore, combining the EGNN [1] model with PIORF appears feasible and presents an intriguing direction for future research. The future research are added to Section 7.
>
>  > [1] Satorras et al. "E(n)-equivariant Graph Neural Networks." ICML, 2021.
>
>
>
> **Q3. Could the approach be applied together with constraint-aware graph simulators like C-GNS?**
>
> Our model basically uses the core framework of GNS[1] and MGN[2]. C-GNS[3] is used in combination with these frameworks. Since PIORF can be applied without changing the model, it appears that it can be applied together.
>
> Since GNS, the core framework used in C-GNS, is the same as the framework used in MGN, it seems feasible to apply PIORF alongside them without modifying the model. However, it may be necessary to define a constraint function that is specifically suited for fluid dynamics
>
> > [1] Alvaro Sanchez-Gonzalez, et al. "Learning to simulate complex physics with graph networks." ICML, 2020
> > [2] Tobias Pfaff, et al. "Learning Mesh-Based Simulation with Graph Networks." ICLR, 2020.
> > [3] Yulia Rubanova, et al. "Constraint-based graph network simulator". ICML, 2022
>
> **Q4. Are there any real-world dataset on which the model can be evaluated? Practitioners would be more exciting to see how the method performs in actual real-world scenarios.**
>
> Please refer to W3.

---

> ### Author Response · Authors · 2024-11-24
>
> Dear Reviewer isvf,
>
> We appreciate your positive ratings and constructive feedback, which have helped us strengthen our paper. We are giving you a reminder as there are only a few days left until the discussion period ends. We have carefully considered your valuable suggestions to further improve our paper. We will appreciate your continued support if you are satisfied that our responses and revisions have addressed your concerns and comments. If you have any additional suggestions for improvement, please let us know.
>
> Best,
>
> Authors

---

> > ### Comment · Reviewer_isvf · 2024-11-26
> >
> > Thank you for the rebuttal. I remain positive on the paper. Regarding W2, my point is that these quantities would be better to be explained in the main text since they are directly related to the understanding of the experiment results. Instead of looking into the appendix, this would offer the readers clearer view.

---

> ### Author Response · Authors · 2024-11-27
>
> We totally agree with you and have added the relevant content to the 'Physical interpretation' paragraph in Section 4, highlighting it in red.

---

### Official Review · Reviewer_Jfxr · 2024-11-03

**Soundness:** 3
**Presentation:** 4
**Contribution:** 3
**Rating:** 8
**Confidence:** 5

**Summary:**

The paper proposes a novel graph rewiring method when using GNNs to solve PDEs. The basic idea is to combine regions based on both graph topology (Ollivier-Ricci curvature (ORC)) and physical features defined on each graph node (e.g. velocity). The proposed method first identifies bottleneck regions, i.e., with low ORC at the nodes, and then connects them with nodes having a maximum difference in physical properties. It seems that the first step identifies well-connected graph clusters typically far away on the graph and then connects them through basically one or two other nodes, namely the one(s) having the lowest and/or highest chosen physical feature. This way, information can propagate quickly between distant regions. The presented experiments seem promising.

**Strengths:**

- Combining topological and physical features. ORC seems meaningful for identifying clusters. Combining it with the physical features also makes sense, as these are quantities of interest.
- Performance on the presented benchmarks is good.
- Overall, the manuscript is well written and has many ablations/baselines.

**Weaknesses:**

- **W1: Why ORC?** I get the point of wanting to connect clusters (according to the used library for computing the ORC, low ORC reveals "bridges within clusters": https://pypi.org/project/GraphRicciCurvature/), but I'm not convinced that this is the best way. Note that I don't say that it is not the right way, I just say that there is no discussion of alternatives.
- **W2: Why not node degree?** If the ORC and the degree of a node are strongly correlated, there should not be much difference between using the degree of a node versus a low ORC. I strongly recommend ablating this design choice (**Ablation 1**). If both perform comparably, there is no reason to use ORC, as the degree of a node is a much simpler concept. Thus, Occam's razor would favor it, and talking about adaptation by other researchers, one would not have to use a separate library to compute curvatures.
- **W3: 6.4 Ablations.** I was not expecting this amount of ablations regarding the second half of your algorithm (lines 5-6 of Algorithm 1, physical features), but I was definitely expecting more ablations regarding the first part (lines 3-4 of Algorithm 1, ORC). In particular, how about randomly picking $S$ nodes instead of the current lines 5-6 of the algorithm relying on the ORC (**Ablation 2**)? This would show the importance of using physical features for rewiring, which seems to be a major part of the approach.
- **W4: Simplest baseline.** I appreciate your effort in the model comparison part! However, I'm missing the simplest possible rewiring technique: randomly pick the same number of bidirectional edges as you use ($S$ if I'm not missing something), and see how this model performs (**Ablation 3**). From what I know, the chosen baselines (DIGL, SDRF, etc.) are designed for very different downstream tasks, and I'm not surprised that they underperform.

I'll be more than happy to increase my score upon adding the suggested Ablations 1-3.

**Minor/Typos:**
- L. 376: "changes" -> "change"?
- L. 425: "boxe" -> "boxes"?
- L. 462:  "PIORF maintains the lowest computation time in all datasets and edge counts." Is this true? The orange FoSR line is nearly identical; by the way, it is also pretty hard to see the orange line, probably even impossible if I print the manuscript on paper -> please think of a better visualization (e.g. log y-axis?). Please fix the whole paragraph regarding which approach is "the fastest", as there seem to be two of them.

**Questions:**

See weaknesses.

---

> ### Author Response · Authors · 2024-11-22
>
> We appreciate Reviewer Jfxr for the positive feedback and insightful comment, highlighting our strengths in
> * Combining topological and physical features.
> * the manuscript is well written and has many ablations/baselines.
>
> Below, we carefully address your concerns. We uploaded the revised paper, where changes are highlighted in red.
>
>
> **W1. Why ORC? I get the point of wanting to connect clusters (according to the used [library](https://pypi.org/project/GraphRicciCurvature/) for computing the ORC, low ORC reveals "bridges within clusters":), but I'm not convinced that this is the best way. Note that I don't say that it is not the right way, I just say that there is no discussion of alternatives.**
>
>
> The reason we use ORC is that it not only captures the nodes and edges near the boundary conditions that alter fluid flow from a domain knowledge, but also identifies potential bottlenecks created by these nodes and edges. Metrics that can be presented as alternatives with a similar role to ORC are Forman curvature [1], Ricci Flow [2], and Betweenness Centrality [3]. The discussion of alternatives are added to Appendix A.
>
> The problem of considering inter-cluster connections can be seen as an important metric in fields involving multiple objects, such as multibody dynamics, a hierarchical model [4] that message passing at two different resolutions (fine and coarse). The rewiring method that considers connectivity between clusters is added in Section 7 as a future research topic.
>
> > [1] RPS reejith et al. "Forman curvature for complex networks" Journal of Statistical Mechanics. 2016.
> > [2] Chien-Chun Ni. "Community Detection on Networks with Ricci Flow" Scientific Reports 2019.
> > [3] M. Barthélemy. "Betweenness centrality in large complex networks" The European Physical Journal B.
> > [4] Meire Fortunato et al. "MultiScale MeshGraphNets." ICML 2022 2nd AI for Science Workshop. 2022.
>
>
>
> **W2~W4 Additional Ablation Study.**
>
> We address W2 to W4 by integrating them into a single response. Our proposed rewiring method includes node selection steps that depend on factors such as degree, ORC, and physical context. To evaluate performance across different factor selections, we conducted additional ablation studies.
>
> Table shows performance based on ingredient selection in CylinderFlow. The first step is to select nodes based on curvature(**"Former"**, Algorithm 1 lines 3-4), and the second is to select nodes based on physical context(**"Latter"**, Algorithm 1 lines 5-6).
>
> **"Ablation 1: effect of high degree"** shows results with high-degree selection, achieving improved performance compared to MGN. However, it underperforms relative to PIORF, as it exhibits varying curvature values for the same degree. **"Ablation 2: physical features"** and **"Ablation 4: graph topology"** show performance based on the choice of physical context and ORC, respectively. Both outperform MGN, and Physical Context provides slightly better performance than ORC. **"Ablation 3: randomly"** is the result of randomly selecting both the former and the latter and adding edges, and is similar to the performance of MGN.
>
> The results of additional experiments are added to Appendix F.
>
> |   Method   | Former |      Latter      | Velocity RMSE | Pressure RMSE |
> |:----------:|:------:|:----------------:|:-------------:|:-------------:|
> |  MGN       |        |                  |      48.8     |      36.7     |
> |    PIORF   |   ORC  | Physical context |      41.6     |      28.9     |
> | Ablation 1 | Degree | Physical context |      44.9     |      33.3     |
> | Ablation 2 | Random | Physical context |      44.6     |      31.1     |
> | Ablation 3 | Random |      Random      |      48.4     |      35.3     |
> | Ablation 4 |   ORC  |      Random      |      43.4     |      32.3     |
>
>
> **Minor/Typos**
>
> Thank you for your valuable feedback. Errors and log scale were changed in the uploaded paper

---

> > ### Comment · Reviewer_Jfxr · 2024-11-26
> >
> > Thanks for the additional ablations! I really like the added table -> I increase my score by 1 point.
> >
> > Yet **W1** still is something I wish I knew more about, rather than mentioning it in future work.

---

> > > ### Author Response · Authors · 2024-11-28
> > >
> > > We would like to clarify W1 as follows.
> > >
> > > We notice that you mentioned how the referenced library uses ORC where “low ORC reveals bridges within clusters.” However, there’s an important distinction in how we use ORC in PIORF. While that library uses ORC for community detection by identifying the lowest connections between clusters, PIORF extends ORC to node-level calculations for selecting sender nodes and identifying bottlenecks in mesh structures for rewiring.
> > >
> > > *If you’re specifically interested in alternatives for selecting sender nodes in PIORF*, we carefully considered 2 main alternatives:
> > > 1. Forman curvature: While computationally more efficient, it is “less geometrical” as noted in Chien-Chun, et al. (page 4) [1]. We chose ORC specifically because it better captures geometric characteristic, particularly around boundary conditions where fluid flow changes dramatically.
> > > 2. Betweenness centrality: While this could be used for source node selection, its high computational complexity ($\mathcal{O}(|\mathcal{V}||\mathcal{E}|)$) and need for global information make it impractical for mesh graphs with thousands of nodes and edges.
> > >
> > > In studies proposing rewiring methods in the field of graph machine learning, metrics with various curvature concepts (i.e., ORC and Forman curvature) are typically used to optimize edge addition or removal by maximizing their values[2-3]. In physics simulation, alternatives to our ORC-based approach could include methods like SDRF and BORF that use different metrics for optimization. For instance, SDRF uses balanced Forman curvature, which provides a more conservative estimate compared to ORC (as shown in Lemma 4.1 of Nguyen et al. [2]). While BORF similarly uses ORC for rewiring, our experiments in Table 1 demonstrate why it’s less suitable for physics simulation. Here, we note that our method is used only for the purpose of selecting sender nodes by extending the value of ORC to nodes.
> > >
> > >
> > > We have included this detailed discussion in Appendix G.4, and we appreciate your question, as it helped us strengthen our justification for using ORC through these  comparisons.
> > >
> > > > [1] Chien-Chun Ni et al., "Community Detection on Networks with Ricci Flow". Scientific reports, 2019
> > > > [2] Khang Nguyen et al., "Revisiting over-smoothing and over-squashing using ollivier-ricci curvature". ICML, 2023
> > > > [3] Jake Topping et al. "Understanding over-squashing and bottlenecks on graphs via curvature" ICLR, 2022

---

> > > > ### Comment · Reviewer_Jfxr · 2024-12-03
> > > >
> > > > Sorry for the last-minute reply; I've been sick for the past 5 days.
> > > >
> > > > Adding this discussion on "why ORC" is what I was missing. With this addition, I think it is a well-rounded story. Thanks

---

> ### Author Response · Authors · 2024-11-24
>
> Dear Reviewer Jfxr,
>
> Thank you for your thoughtful and detailed comments, which have helped us enhance our paper. As we have only a few days until the discussion period ends, we wanted to remind you of our responses to your valuable feedback.
>
> We have carefully implemented and analyzed "ablation studies of W2-4" as you recommended, and the results and discussions are in our response.  Given that we have addressed this feedback, we hope you will consider increasing your score as you indicated. If you have any additional thoughts about our ablation studies or other aspects, we would be happy to address them.
>
> Best regards,
>
> Authors

---

> ### Author Response · Authors · 2024-12-01
>
> Dear Reviewer Jfxr,
>
> Regarding your question about **W1**, we would like to check if our explanation has adequately addressed your concerns. As our discussion period has been extended **until December 2nd*, we remain open to further discussion if you have any additional questions or points you would like to explore. We appreciate our discussion so far and are happy to continue clarifying any aspects of our work.
>
> Best regards,
>
> Authors

---

### Official Review · Reviewer_5Uy1 · 2024-11-04

**Soundness:** 2
**Presentation:** 2
**Contribution:** 2
**Rating:** 5
**Confidence:** 4

**Summary:**

In this paper, authors propose Physics-Informed Ollivier–Ricci Flow (PIORF) which builds on the Ollivier–Ricci Flow. This innovative rewiring method integrates physical correlations with graph topology to address the over-squashing problem, which traditional approaches often overlook by focusing solely on graph topology without considering underlying physical phenomena. The proposed method is designed with the following 3 goals: 1) physical context, 2) efficiency, 3) accuracy. Furthermore, authors extend the ORC to node-level curvature, which is the core of the RIOFR. Experimental results on 3 fluid dynamics benchmark datasets show that PIORF consistently outperforms baseline models and existing rewiring methods.

**Strengths:**

1. The authors introduce a novel Ollivier–Ricci Flow, termed PIORF, to address the over-squashing problem that neglects physical phenomena.

2. The authors extend the ORC to node-level curvature.

3. The effectiveness of the method is validated through experimental verification.

**Weaknesses:**

1. There is a lack of sufficient theoretical analysis compared with former works.

2. The paper does not provide an explanation for how the added edges address the underlying physical phenomena.

3. The legends in the figures are not clear.

**Questions:**

1(W1). Inadequate theory compared to previous work(BORF)[1]. And authors complain that 'BORF works in batches and calculates the curvature with a minimum and maximum in each batch. Then, connections are added to the set with the minimum edge value to uniformly weaken the graph bottleneck. To save computation time, BORF does not recalculate the graph curvature within each batch, but rather reuses the already computed optimal transfer plan between sets to determine which edges should be added.', but there is no indication in this paper that there is an increase in efficiency compared to BORF

2(W1). The addition of edges also changes the topology, which may lead to negative consequences that are not addressed in the paper.

3(W1). Computational efficiency is demonstrated only through experimental illustrations of efficiency gains, with no accompanying theoretical analysis.

4(W1). Contrary to the existing theory. Topping's work [2] suggests that a fully connected graph does not have over-squashing (maximum curvature). However, as shown in Figure 3(a), the authors claim that the degree of a node is negatively correlated with curvature. These two theories are clearly contradictory. The authors should clarify whether this discrepancy arises from specific conditions discussed in the context of fluid dynamics. The degree may not be correlated to the level of curvature.

5(W2). There is no clear rationale for why adding edges between $s$ and $r$ would address the underlying physical phenomena. The authors only provide some intuitive insights from physics and experimental simulations, but these edges could potentially have a negative impact.

6(W3). Legends should be provided, and the meanings of the colors should be clarified to help readers better understand the impact of color changes in the figures (Figure 1, 4, 7, 8, 9,10 11).

[1] Khang Nguyen, Nong Minh Hieu, Vinh Duc Nguyen, Nhat Ho, Stanley Osher, and Tan Minh Nguyen. Revisiting over-smoothing and over-squashing using ollivier-ricci curvature. In In- ternational Conference on Machine Learning, pp. 25956–25979. PMLR, 2023

[2] Jake Topping, Francesco Di Giovanni, Benjamin Paul Chamberlain, Xiaowen Dong, an Michael M Bronstein. Understanding over-squashing and bottlenecks on graphs via curvature. arXiv preprint arXiv:2111.14522, 2021

---

> ### Author Response · Authors · 2024-11-22
>
> **Q1. (W1). Inadequate theory compared to previous work(BORF). And authors complain that 'BORF works in batches and calculates the curvature with a minimum and maximum in each batch. Then, connections are added to the set with the minimum edge value to uniformly weaken the graph bottleneck. To save computation time, BORF does not recalculate the graph curvature within each batch, but rather reuses the already computed optimal transfer plan between sets to determine which edges should be added.', but there is no indication in this paper that there is an increase in efficiency compared to BORF**
>
> Our paper focuses on computational aspects of the MGN method; the main goal is to develop novel rewiring mechanism, which considers physical quantities of complex physical phenomena, for improved computational efficiency. To this end, our experimental design is oriented toward large-scale and complex physical simulations. We make comparisons against other rewiring algorithms in extensive sets of experiments. Although the proposed PIORF is less equipped with theories, we attempted to support the effectiveness of the methods by providing strong empirical evidences.
>
> To iterate the difference between BORF and the proposed PIORF, BORF uses rewiring batches, which increases computational cost with each batch iteration. Figure 6 in Section 6.3 shows the computation times for several baselines and confirms that BORF is the slowest. In addition, BORF has three hyperparameters: the number of rewiring batches, edge addition, and edge removal, which can be a disadvantage when trying to find the optimal values. In contrast, PIORF has only one hyperparameter: the pooling ratio.
>
>
> **Q2. (W1). The addition of edges also changes the topology, which may lead to negative consequences that are not addressed in the paper.**
>
> We thank your concern about the potential negative consequences of topology changes. Our PIORF is carefully designed to add only meaningful connections that enhance information flow while preserving physical relationships in the mesh. Let us explain this more precisely:
>
> Our edge addition follows two principles:
> *  We identify topological bottleneck regions using Ollivier-Ricci curvature
> *  We connect nodes with significant velocity differences to capture important physical interactions
>
> To theoretically show that these additional edges improve rather than harm the ability to propagate information, we analyze the effective resistance between nodes. Following [1], effective resistance measures how well information can flow between nodes $u$ and $v$ in a graph:
> $$R_{u,v} = \sum_{i=0}^{\infty} (\frac{1}{d_u} \hat{A}^i_{uu}  + \frac{1}{d_v}  \hat{A}^i_{vv}  - \frac{2}{\sqrt{d_u d_v}} \hat{A}^i_{uv} ), $$ where $(\hat{A}^{i})_{u,v}$ represents the number of paths of length $i$ between nodes $u$ and $v$.
>
> This equation shows that effective resistance decreases with more and shorter paths between nodes.
> Our empirical analysis supprots that PIORF reduces the total effective resistance ($R_{tot} = \frac{1}{2} \sum_{u,v} R_{u,v}$). We randomly sampled 100 nodes from each dataset and measured the total effective resistance before and after applying PIORF:
> - CylinderFlow: Reduction from 2,433,015 to 1,606,768
> - Airfoil: Reduction from 15,644,891 to 10,143,689
>
> These reductions (34% and 35%, respectively) show that our method effectively alleviates topological bottlenecks by adding physically meaningful connections. We believe that we address the reviewer's concern about potential negative consequences.
>
> > [1] Mitchell Black et al. "Understanding Oversquashing in GNNs through the Lens of Effective Resistance". ICML, 2023.
>
> **Q3. (W1). Computational efficiency is demonstrated only through experimental illustrations of efficiency gains, with no accompanying theoretical analysis**
>
> The complexity of PIORF is $\mathcal{O}(|\mathcal{E}|d^3_{\mathrm{max}})$, where $|\mathcal{E}|$ is the number of edges and $d_\mathrm{max}$ is the maximal degree. Since PIORF does not rely on batch or iterative computations, it is particularly effective for datasets with a large number of nodes, such as those in fluid dynamics or mechanics. The complexity is added to Appendix A and highlighted in red.

---

> ### Author Response · Authors · 2024-11-22
>
> **Q4. (W1). Contrary to the existing theory. Topping's work suggests that a fully connected graph does not have over-squashing (maximum curvature). However, as shown in Figure 3(a), the authors claim that the degree of a node is negatively correlated with curvature. These two theories are clearly contradictory. The authors should clarify whether this discrepancy arises from specific conditions discussed in the context of fluid dynamics. The degree may not be correlated to the level of curvature.**
>
> The ratio of negative edge curvature in CylinderFlow is 21.88%. These negative values can potentially cause bottlenecks. We converted this edge curvature to node curvature by averaging with neighboring nodes and analyzed the relationship between degrees, confirming that higher degrees correspond to higher negative curvature.
>
> According to Deac et al., 2022[1], nodes with high degrees can cause bottlenecks, thereby exacerbating the issue of over-squashing. In fluid mechanics, which primarily uses triangular meshes, non-uniform degree distributions form near boundary conditions such as walls, resulting in nodes with high degrees.
>
> > [1] Andreea Deac et al. "Expander graph propagation". NeurIPS Workshop, 2022
>
>
> **Q5. (W2). There is no clear rationale for why adding edges between $s$ and $r$ would address the underlying physical phenomena. The authors only provide some intuitive insights from physics and experimental simulations, but these edges could potentially have a negative impact.**
>
>
> In fluid dynamics, governing equations such as Navier Stokes are helpful in predicting fluid flow because they depend on gradients of velocity[1]. the distinction between laminar and turbulent flows, as quantified by velocity and the Reynolds numbers, is important for understanding system behavior.
>
> Velocity gradients are an important element in dynamics and have an equally important influence on learning. We confirmed that the accumulated error is related to the velocity gradient, and the result of discussion is added to Appendix G.3.
>
> PIORF effectively integrates this physical context into graph rewiring by adding edges between nodes with significant velocity differences. This allows the model to help with long-range interactions to better simulate real-world phenomena such as fluid turbulence. To confirm the effect of physical phenomena, we conduct additional experiments by randomly selecting nodes and applying physical context.
>
> Table is the result before and after rewiring and shows better performance after applying the physical context in CylinderFlow. The results of additional experiments are added to Appendix F.
>
> |   Method   | Velocity RMSE | Pressure RMSE |
> |:----------:|:-------------:|:-------------:|
> |  Original  | 48.8          |      36.7     |
> |  Physical context   | 44.6          |      31.1     |
>
> >[1] Charles Meneveau. "Lagrangian dynamics and models of the velocity gradient tensor in turbulent flows." Annual Review of Fluid Mechanics, 43(1):219–245, 2011.
>
>
> **Q6. (W3). Legends should be provided, and the meanings of the colors should be clarified to help readers better understand the impact of color changes in the figures (Figure 1, 4, 7, 8, 9,10 11).**
>
> Thank you for your valuable comments, we have added legends to all figures in the uploaded paper.

---

> ### Author Response · Authors · 2024-11-24
>
> Dear Reviewer 5Uy1,
>
> We appreciate your comments on helping us improve our paper in many aspects. This is a gentle reminder since we have only a few days until the discussion period ends. We have tried our best to address your questions. We would appreciate it if you could either confirm that our responses have satisfactorily addressed your concerns and consider a more favorable score or let us know if you have any follow-up questions.
>
> Best,
>
> Authors

---

> > ### Comment · Reviewer_5Uy1 · 2024-11-25
> >
> > Q1. The author demonstrated the effectiveness of the work through experiments and practical experience, but theoretical support is important.
> >
> > Q2. The method proposed by the author effectively alleviates the topological bottleneck by adding physically meaningful connections. However, we are concerned about whether modifying the graph structure by adding edges could influence the experimental outcomes. Specifically, if two graphs are very similar, the process of adding edges might result in two identical graphs, potentially impacting the results. This raises a question: since the original graphs are different, their results based on the original structure could also differ, possibly contradicting the final outcomes. Therefore, we request that the author clarify whether this scenario could lead to two originally distinct graphs producing the same result.
> >
> > Q4. I didn't find this conclusion in the original paper[1]. I just found that the curvature of the node with the highest degree of nodes is non-negative.
> > [1] Andreea Deac et al. "Expander graph propagation". NeurIPS Workshop, 2022

---

> > > ### Author Response · Authors · 2024-11-27
> > >
> > > We are grateful for your assistance and constructive feedback. Your insights have been invaluable in improving the quality.
> > >
> > > The field of learning dynamics [1-4] holds significant potential for enhancing the analysis speed of traditional tools used by practitioners and for replacing these tools by improving performance. We would like to emphasize that existing methods are not well-suited for this field and that rewiring, which incorporates physical quantities based on domain knowledge, represents a novel approach. We believe that our findings will greatly benefit both the learning dynamics and mesh graph communities.
> > >
> > > > [1] Alvaro Sanchez-Gonzalez et al. "Learning to Simulate Complex Physics with Graph Networks" ICML 2020
> > > > [2] Tobias Pfaff et al. "Learning Mesh-Based Simulation with Graph Networks" ICLR 2021
> > > > [3] Yadi Cao et al. "Efficient Learning of Mesh-Based Physical Simulation with Bi-Stride Multi-Scale Graph Neural Network" ICML 2023
> > > > [4] Steeven Janny et al. "EAGLE: Large-scale Learning of Turbulent Fluid Dynamics with Mesh Transformers" ICLR 2023
> > >
> > > **Q1. The author demonstrated the effectiveness of the work through experiments and practical experience, but theoretical support is important.**
> > >
> > > We agree that theoretical support is important. While existing rewiring papers have addressed a significant amount of theoretical content, it is equally crucial to experimentally demonstrate that rewiring methods considering physical factors are more effective than existing methods in various learning dynamics models, such as MGN, BSMS, and HMT.
> > >
> > >
> > > **Q2. The method proposed by the author effectively alleviates the topological bottleneck by adding physically meaningful connections. However, we are concerned about whether modifying the graph structure by adding edges could influence the experimental outcomes. Specifically, if two graphs are very similar, the process of adding edges might result in two identical graphs, potentially impacting the results. This raises a question: since the original graphs are different, their results based on the original structure could also differ, possibly contradicting the final outcomes. Therefore, we request that the author clarify whether this scenario could lead to two originally distinct graphs producing the same result.**
> > >
> > > Particularly in graph-level tasks, we somewhat recognize the reviewer’s concern about similar graphs potentially becoming identical through edge addition. This issue can be critical in domains such as molecular graphs or social networks, where modifying the graph structure by adding or removing edges may violate domain constraints and specific structural meaning [1,2].
> > >
> > > We would like to clarify that this concern does not apply to our mesh-based simulation datasets (CylinderFlow and AirFoil) for several key reasons. These datasets represent physical simulations where each trajectory maintains the same mesh graph throughout its evolution. The social or molecular graphs whose topology has semantic meaning would have problems if two other graphs had the same graph structure due to added edges. However, this does not necessarily happen in our task. For example, consider the CylinderFlow dataset containing 1,000 trajectories with 600 timesteps each. While different trajectories may have varying cylinder positions/sizes, resulting in distinct mesh structures, within each trajectory, the mesh graph structure remains constant. You can refer to the simulation videos (https://sites.google.com/view/meshgraphnets).
> > >
> > > Following the reviewer’s point, if graphs at steps t-1 and t are identical, and if PIORF adds the same edges at these two steps, this would indicate an equilibrium state where there is minimal velocity change between steps. This actually uses domain knowledge about fluid flow velocities and similar properties, thus avoiding negative impacts. In fact, existing rewiring methods commonly used in graph-level tasks might potentially have negative effects as they don’t consider simulation-specific requirements, which is clearly shown as not always improved performances in Table 1.
> > >
> > > > [1] Jeongwhan Choi et al. "PANDA: Expanded Width-Aware Message Passing Beyond Rewiring" ICML 2024
> > > > [2] Mengying Sun et al. "MoCL: Data-driven Molecular Fingerprint via Knowledge-aware Contrastive Learning from Molecular Graph" KDD 2021

---

> > > ### Author Response · Authors · 2024-11-27
> > >
> > > **Q4. I didn't find this conclusion in the original paper[1]. I just found that the curvature of the node with the highest degree of nodes is non-negative.**
> > > The authors [1] state that such features are insufficient for sufficiently large graphs and suggests that limiting the degree is useful when the graph size is large. Since the graph we use is a large graph with over 10,000 edges, this approach can help alleviate over-squashing. Furthermore, according to this paper [2], removing high-degree nodes leads to better performance.
> > >
> > > > [1] Andreea Deac et al. "Expander graph propagation". NeurIPS Workshop, 2022
> > > > [2] Hugo Attali et al. "Delaunay Graph: Addressing Over-Squashing and Over-Smoothing Using Delaunay Triangulation" ICML, 2024

---

> ### Comment · Reviewer_5Uy1 · 2024-11-27
>
> Thank you very much for the explanation and effort in the rebuttal period.
>
> Q4. I didn't find that conclusion in paper [1]. In paper [2], the only relevant statement I encountered was that 'High-degree nodes are particularly prone to over-squashing'. This does not necessarily imply that the improved results are due to alleviating over-squashing by removing the heigh degree nodes. It’s possible that removing the node with the high degree has led to other effects that enhance performance.

---

> > ### Author Response · Authors · 2024-11-27
> >
> > **Q4. I didn't find that conclusion in paper [1]. In paper [2], the only relevant statement I encountered was that 'High-degree nodes are particularly prone to over-squashing'. This does not necessarily imply that the improved results are due to alleviating over-squashing by removing the heigh degree nodes. It’s possible that removing the node with the high degree has led to other effects that enhance performance.**
> >
> > Thank you for the opportunity to address the reviewer's follow-up question regarding Q4. We appreciate the reviewer's careful examination of the references and would like to clarify our position with a more fine analysis. We also agree with your perspective regarding the papers [1-2] and would like to highlight a situation that occurs exclusively in mesh graphs.
> >
> > First, we would like to address your previous Q4. While Topping et al. [3] show that fully connected graphs do not show over-squashing due to maximum curvature, Triangular mesh graphs we used show a more complex case. Though they don't strictly fall into any of the three categories shown in Figure 2 of Topping et al. [3], they share characteristics with grid networks but with important differences.
> >
> > In general, the mesh structure shows a planar geometry similar to that of grid networks in most regions. However, near boundary conditions and in areas with mesh refinement, the curvature properties can vary significantly. Our empirical analysis indicates that nodes with high degrees can introduce areas of negative curvature and partially show characteristics of tree-like hyperbolic geometry. This means our mesh graphs are hybrid structures - mostly planar but with localized regions that can show hyperbolic characteristics, particularly around high-degree nodes created by mesh refinement or boundary conditions. By analyzing these features in the mesh, as shown in Figure 3a in Section 5, we empirically confirmed that the mesh graph used in the field of fluid dynamics exhibits the creation of negative curvature around high-degree nodes.
> >
> > > [1] Andreea Deac et al. "Expander graph propagation". NeurIPS Workshop, 2022
> > > [2] Hugo Attali et al. "Delaunay Graph: Addressing Over-Squashing and Over-Smoothing Using Delaunay Triangulation" ICML, 2024
> > > [3] Jake Topping et al. "Understanding over-squashing and bottlenecks on graphs via curvature" ICLR, 2022

---

> ### Author Response · Authors · 2024-12-01
>
> Dear Reviewer 5Uy1,
>
> We noticed that you raised your initial rating without comment and we appreciate you recognizing our efforts. However, we are writing to follow up on our response to **Q4** and note that the discussion period has been extended **until December 2nd**.
>
> If you have had the chance to review our response to Q4, we believe our explanation provides important technical insights about the geometric properties of mesh graphs in fluid dynamics applications. ***We would like to understand if there are any remaining concerns preventing an acceptance score and what the key reasons are for your score remaining below the acceptance bar.***
>
> Additionally, in our response to Q2, we provided a theory-based intuitive explanation of how PIORF's appropriate edge addition can mitigate over-squashing from an effective resistance perspective while also addressing your concerns about the potential negative impacts of edge addition. In Q3, we discussed computational efficiency with theoretical analysis, and in Q4, we explained why we designed our rewiring algorithm using physical context.
>
> Thanks to your valuable feedback, we improved our paper significantly. We remain open to further discussion if you have any remaining questions. Thank you for your time and thoughtful engagement in this review process.
>
> Best regards,
>
> The Authors

---

### Meta-Review · Area_Chair_FTSp · 2024-12-14

**Metareview:**

This paper introduces a novel graph rewiring method called Physics-Informed Ollivier–Ricci Flow (PIORF) to address the over-squashing problem in graph neural networks (GNNs) applied to fluid dynamics simulations. PIORF extends the Ollivier–Ricci curvature (ORC) to node-level curvature by integrating velocity gradients. It can identifies bottleneck regions in mesh-based graphs and connects them through nodes with significant physical differences, enhancing long-range interactions.

The idea of the combination of graph graph topology and physical features to mitigate over-squashing in GNNs for fluid simulations is novel. The experiment is sufficient and well organized, with convincing results and comprehensive ablation studies. However, the theoretical justifications looks insufficient. The impact of over-squashing in fluid simulations is underexplored. It is uncertain whether this is the primary factor contributing to performance decline.

Considering the innovative approach, solid experimental results, and the potential impact on improving GNN performance in fluid dynamics simulations, I recommend the acceptance of this paper,  and the authors are encouraged to address the highlighted concerns from Reviewers to strengthen the final version.

**Additional Comments On Reviewer Discussion:**

During the rebuttal period, everyone had participated the discussion and acknowledge  the responses made by  authors.

---

### Decision · Program_Chairs · 2025-01-22

Accept (Poster)